**Subject Category:**
Biology (whole organism)

palaeontology/taxonomy and systematics/evolution

Carboniferous, Mazon Creek, Eureptilia, Parareptilia, Amniota, Pennsylvanian

**Author for correspondence:**
Arjan Mann
e-mail: arjan.mann@carleton.ca

# Carbonodraco lundi gen et sp. nov., the oldest parareptile, from Linton, Ohio, and new insights into the early radiation of reptiles

Arjan Mann, Emily J. McDaniel, Emily R. McColville and Hillary C. Maddin

Department of Earth Sciences, Carleton University, 1125 Colonel By Drive, Ottawa, Ontario, Canada K1S 5B6

AM, 0000-0002-3511-0635; HCM, 0000-0002-6969-4907

Redescription of the holotype specimen of *Cephalerpeton ventriarmatum* Moodie, 1912, from the Middle Pennsylvanian (Moscovian) Francis Creek Shale of Mazon Creek, Illinois, confirms that it is a basal eureptile with close postcranial similarities to other protorothyridids, such as *Anthracodromeus* and *Paleothyris*. The skull is long and lightly built, with large orbits and a dorsoventrally short mandible similar to most basal eureptiles. Two specimens referred previously to *Cephalerpeton* cf. *C. ventriarmatum* from the approximately coeval Linton, Ohio, locality differ significantly from the holotype in cranial and mandibular proportions and tooth morphology. This material and an additional Linton specimen compare favourably to 'short-faced' parareptiles, such as *Colobomycter* and *Acleistorhinus*, and justify recognition of an aleistorhinid parareptile in the Linton assemblage. The new binomen is thus the oldest known parareptile.

## 1. Introduction

Amniotes can be divided into two major lineages, synapsids (mammals and their extinct relatives) and reptiles (crocodiles, birds, and lepidosaurs, and their extinct relatives). The origin and early diversification of these groups are believed to have occurred during the early Carboniferous because the oldest amniotes, the reptile *Hylonomus lyelli* and the putative synapsid *Protoclepsydrops haplous*, are known from the classic locality of Joggins, Nova Scotia, Canada (313–316 Ma). Other early records of non-synapsid amniotes are from the Coal Measure localities of

Mazon Creek, Illinois, and Linton, Ohio, USA (*Cephalerpeton* and *Anthracodromeus*); Florence, Nova Scotia (*Paleothyris*); and Nýřany, Czech Republic (*Brouffia* and *Coelostegus*). These taxa were regarded by Carroll & Baird [1] as stem-reptiles within the now-defunct family Romeriidae and were assigned subsequently to the Protorothyrididae in a number of papers [2–4]. The most recent phylogenetic analysis of these taxa found the Protorothyrididae to be an array of basal eureptiles and thus paraphyletic [5]. Yet, these same authors recognized a clade consisting of *Cephalerpeton*, *Anthracodromeus*, and the Lower Permian *Protorothyris* from North-Central Texas as a sister taxon of diapsids.

*Cephalerpeton ventriarmatum* Moodie 1912 was described from an articulated anterior half skeleton preserved in a concretion that lacks a counterpart from the Francis Creek Shale of the classic Mazon Creek region. Moodie [6] briefly described this animal as a microsaurian amphibian, which at that time was an amalgamation of small tetrapods. The specimen was described more fully by Gregory [7] and by Carroll & Baird [1]. Reisz & Baird [8] reported additional material of *Cephalerpeton* from Linton, Ohio. These remains consist of a macerated skull (CM 23055) and a mandible (NHMUK R. 2667). Because of perceived differences with the Mazon Creek holotype, Reisz & Baird [8] referred the Linton material to *Cephalerpeton* cf. aff. *C. ventriarmatum*.

Since these descriptive works, *Cephalerpeton* has rarely been included in phylogenetic analysis of reptiles or amniotes despite being important to the origin and evolution of Reptilia. Here, we provide new, comprehensive and comparative descriptions of all specimens that have referred to *Cephalerpeton* from Mazon Creek and Linton. In our study, we were able to study original latex peels and casts of *Cephalerpeton ventriarmatum* from Mazon Creek in order to supplement anatomical analysis of the damaged holotype specimen. We recognize new craniodental features of *Cephalerpeton ventriarmatum* that distinguish it from the Linton material, which we assign to the new acleistorhinid parareptile, *Carbonodraco lundi* gen et sp. nov. As a result, *Carbonodraco lundi* represents the oldest known parareptile.

## 2. Material and methods

Specimens were studied at: American Museum of Natural History (AMNH), New York, New York; Carnegie Museum of Natural History (CM), Pittsburgh, Pennsylvania; Field Museum of Natural History (FMNH), Chicago, Illinois; Redpath Museum at McGill University (RM), Montreal, Quebec; Smithsonian Institution (USNM), Washington D.C.; and Yale Peabody Museum (YPM), New Haven, Connecticut. Additional specimens held by the following institutions were studied on the basis of casts, latex peels or publications: Natural History Museum (formerly British Museum [Natural History]), London (NHMUK); Museum of Comparative Zoology, Cambridge, Massachusetts (MCZ); and Museum für Naturkunde, Humboldt Universität, Berlin (MB). The dataset thus includes nearly all basal eureptiles from the Carboniferous, including notable material from Mazon Creek, Illinois, and Linton, Ohio. We were also able to study material of the following parareptiles: *Delorhynchus*, *Colobomycter*, *Acleistorhinus* and *Erpetonyx* that were on loan to Robert R. Reisz at the University of Toronto Mississauga at the time of this study. A Sony Alpha ILCE 5000 camera with a F3.5 macro lens was used for photography.

Illustration of YPM 796 was redrawn and modified from Carroll & Baird [1], and CM 23055 was redrawn and modified from Reisz & Baird [8]. Anatomical illustrations of CM 81536 and CM 41714 were drawn from original specimens and casts or peels. All figure drawings were generated and formatted in Photoshop CS6 (Adobe, San Jose, CA).

### 2.1. Anatomical abbreviations

**aa**, atlantal arch; **ac**, anterior coracoid; **an**, angular; **ar**, articular; **axa**, axis arch; **axp**, axis pleurocentrum; **bo**, basioccipital; **c**, clavicle; **cth**, cleithrum; **dv**, dorsal vertebrae; **d**, dentary(**rd/ld**, right/left); **ect**, ectopterygoid; **eo**, exoccipital; **f**, frontal; **gs**, gastralia; **h**, humerus; **l**, lacrimal; **j**, jugal; **m**, maxilla; **mc**, metacarpals; **n**, nasal; **p**, parietal; **pf**, postfrontal; **pfo**, pineal foramen; **pmx**, premaxilla; **pp**, postparietal; **prf**, prefrontal; **pl**, palatine; **pt**, pterygoid; **pro**, proatlas; **q**, quadrate; **qj**, quadratojugal; **r**, radius; **s**, scapula; **scl**, scleral ossicles; **sm**, septomaxilla; **so**, supraoccipital; **sq**, squamosal; **st**, stapes; **sv**, sacral vertebra; **sp**, splenial; **t**, tabular; **tfpt**, transverse flange of the pterygoid; **u**, ulna; **v**, vomer.

## 3. Systematic palaeontology

Tetrapoda Jaekel 1909

 Amniota Haeckel 1866

Eureptilia Olson 1947
*Cephalerpeton* Moodie 1912
*Cephalerpeton ventriarmatum* Moodie, 1912

**Hototype:** YPM 796, anterior portion of a skeleton, including the upper limbs, cranium and lower jaws.

**Locality and Horizon:** Mazon Creek, Grundy County, Illinois, U.S.A. Francis Creek Shale, above the Morris (no. 2) Coal, Carbondale Formation, Middle Pennsylvanian (Moscovian).

**Revised Diagnosis:** A basal eureptile diagnosed by the following autapomorphies: 16 wide-based conical teeth in maxilla; maxillary dentition significantly enlarged compared to dentary teeth; maxillary dentition bears multiple peaks in tooth height; palatal bones covered in a shagreen of denticles. Shares a slender, rod-like ulna and radius with *Anthracodromeus*, *Paleothyris*, and basal diapsids.

**Comments:** With the recognition of *Carbonodraco lundi* gen. et sp. nov. (see below), *Cephalerpeton* is known only from YPM 796. One similarity between *Cephalerpeton* and *Carbonodraco* is the presence of plicidentine and intense enamel fluting on the tooth crowns. This trait is also widely shared with other pareptiles and synapsids [9]. Because of its wide dispersal, this trait is not included in the diagnosis, although it may be unique to *Cephalerpeton* among 'protorothyridid' reptiles.

# 4. Description

## 4.1. Cranial anatomy of *Cephalerpeton*

The skull of YPM 796 is crushed and incompletely preserved (figures 1 and 2). Many of the cranial elements are preserved in ventral or medial views and reveal their internal anatomy. As such, dermal ornament of the skull roof cannot be described. Pieces of the palate are disarticulated and poorly represented. Both lower jaws are shifted to the anatomical right side of the cranium. In general, the right side of the cranium is better preserved, showing a nearly complete cheek region that is absent from the left side of the skull. The cranial anatomy of *Cephalerpeton* is overall lightly built, with the elements being thin and with the skull build being quite narrow. This compares well with contemporaneous Carboniferous 'protorothyridids', early captorhinids such as *Euconcordia* [10], as well as diapsids such as *Spinoequalis* [11], *Petrolacosaurus* [4,12] and *Araeoscelis* [13]. Because the premaxillae are not well preserved and because most of the skull table is absent, estimation of the skull length is difficult.

Anteriormost on the skull are preserved remnants of a single left premaxilla. The premaxilla is lightly built and has spaces for at least three teeth. Unlike the reconstruction of Gregory [7], we interpret the premaxilla as bearing a slightly recurved dorsal ascending process, similar to that of other early eureptiles, such as 'protorothyridids' and araeoscelids, but not as hooked as that in captorhinids or recumbirostrans. The dorsal ascending process of the premaxilla has an elongate morphology. The lateral contact between the premaxilla and the maxilla is not preserved in any of the peels or casts examined.

Both maxillae are preserved in medial perspective. The left maxilla is the most completely preserved. There, a thin and delicately built anatomy is seen, similar to that of other basal eureptiles such as *Hylonomus*, *Paleothyris*, *Thuringothyris* and most captorhinids [14,15]. Anteriorly the bone descends to a thin short anterior process that narrowly contributes to the posterior edge of external naris. Just anterior to the orbit, the maxilla ascends to a low facial lamina similar to that of most 'protorothyridid' reptiles. There is a thin, long posterior process that terminates approximately below the posterior margin of the orbit. It appears this area of the maxilla was excluded from the orbit by a point contact between the lacrimal and jugal. The left maxilla preserves 16 tooth positions, and 14 teeth in place. Each tooth consists of a wide cone that bears crenulations on the base and enamel fluting on the crown, and some of the teeth show resorption pits at their bases. The teeth express a greater degree of heterodonty than present in most other early 'protorothyridids'. Although the largest maxillary teeth are located directly under the facial lamina, the largest tooth is located under the anterior margin of the orbit. This means there is no single, distinct area of tooth enlargement or 'caniniform region', but rather at least two peaks in maximum height along the tooth row. There is also no noticeable diminution in tooth height posteriorly as described below in *Carbonodraco*.

The lacrimals are exposed on both sides of the cranium in medial aspect. Their morphology is shared with most 'protorothyridid' reptiles, as well as araeoscelids such as *Petrolacosaurus* [4,12]. Whereas the right lacrimal is obscured by the vomer, the left is nearly perfectly represented. It is a long element

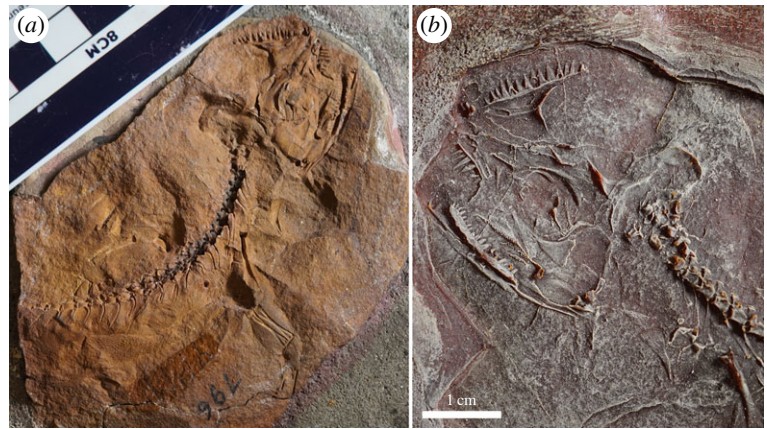

**Figure 1.** *Cephalerpeton ventriarmatum* (YPM 796), (*a*) original nodule (cranium in ventral aspect: postcranium in dorsal lateral aspect) and (*b*) resin cast of the original nodule.

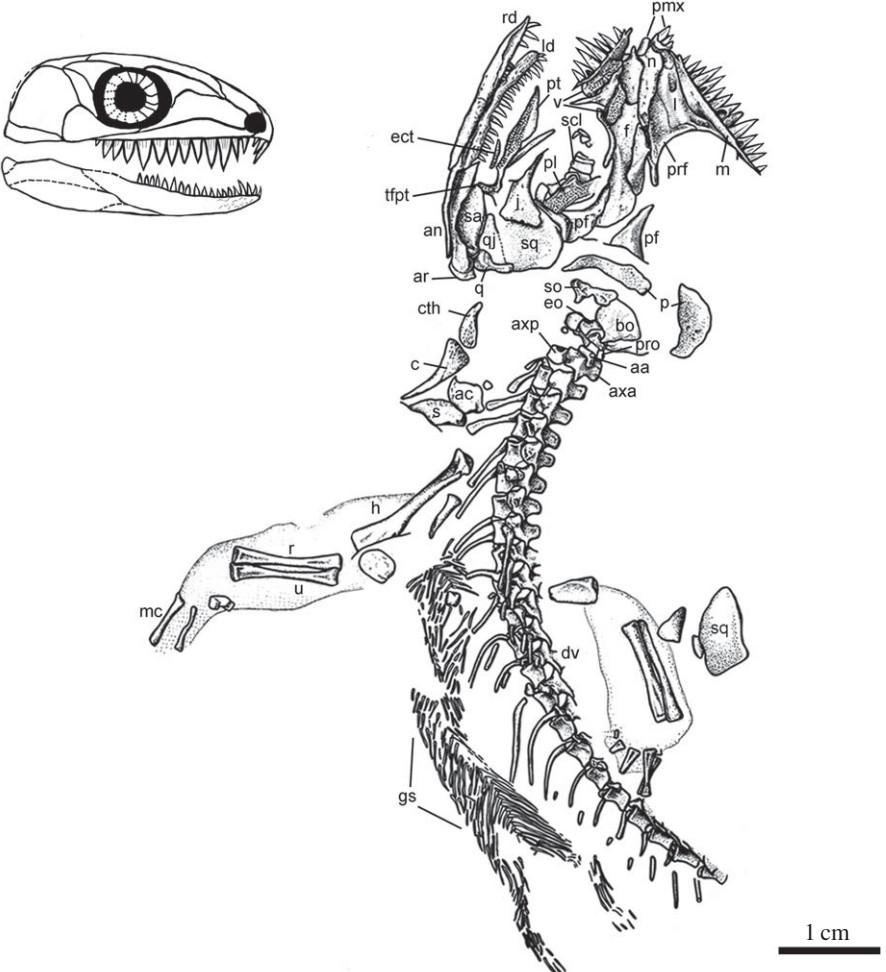

**Figure 2.** Illustration of the cranial and postcranial anatomy of *Cephalerpeton ventriarmatum* (YPM 796), modified from Carroll & Baird [1], and a cranial reconstruction.

that meets the naris anteriorly and contributes to the anterior orbital margin. The left lacrimal also reveals a long lacrimal canal represented by a raised tube terminating at an opening just posterior to the narial opening. The length of the lacrimal canal is significantly longer than that of *Carbonodraco* as described below. Posteriorly on the left lacrimal, there is a recess for the ventrally directed process of the adjoining prefrontal. Both lacrimals preserve a long posteroventrally extending process that meets the anterior process of the jugal, as described by Carroll & Baird [1].

Both prefrontals are present, and the left is more completely exposed. It is excluded from the external naris by a broad contact between the nasal and lacrimal on the lateral surface of the snout. The prefrontal is roughly Y-shaped, with prominent anterior, ventral and posterior processes. These processes are part of a system found on nearly every circumorbital element, a system common among early reptiles, including parareptiles. In *Cephalerpeton*, the anterior process of the prefrontal is extensive, covering a large portion of the lateral snout; the posterior process is quite long and thin, similar to that of other 'protorothyridids'; and the ventrally directed process forms much of the anterior margin of the orbit and fits into a facet on the lacrimal.

The postfrontals are falciform elements preserved in ventral aspect. Each has a somewhat long anterior process and a rather short posterolateral extension onto the cheek. The postfrontal morphology is similar to the condition seen in other basal eureptiles like *Hylonomus* [1], as well as some diapsids such as *Petrolacosaurus* [4].

The nasals are preserved in ventral aspect and have a long, narrow, subrectangular outline. The right nasal slightly overlaps the left; this is a result of taphonomic processes. There is a slight anterior expansion directed toward the narial margin away from the midline. The anteriormost end of the nasals taper to a point and create a midline internasal recess that accommodated the thin ascending dorsal process of the premaxilla.

The frontals are preserved in a similar manner to the nasals in being slightly overlapping, with the right better exposed. They are long rectangular elements that show a small contribution to the dorsal orbital margin. This is indicated by a pinching laterally in mid-region of the frontal. Overall, both the frontals and nasals are extremely narrow, as noted by previous studies [1,7]. The anterior margin of the frontal appears to form a straight sutural contact with the nasal. Because the parietals have shifted out of place and are not well preserved, it is not possible to determine the sutural contact between the frontals and parietals.

The parietals are represented by two large semi-lunate ossifications. Both appear incomplete and possibly would have been better represented on the unknown counterpart. Based on the somewhat short space between the occipital elements and the frontals, it is assumed the parietals were anteroposteriorly short and only moderately expanded lateromedially.

The cheek region is comprised of a jugal, postorbital, squamosal and quadratojugal. These are all represented only on the right side (except for the squamosal), preserved in medial view. Anteriorly, the jugal forms a thin process that contributes to the posteroventral margin of the orbit. This process meets, and possibly slightly overlaps, the lacrimal. The cheek region of the jugal is moderately expanded. There is a small notched contact dorsally where the jugal would have accepted the postorbital and a straight contact posteriorly with the squamosal. The postorbital is not well represented on the right side, where it is overlapped by the squamosal. Taking into account the relationship of the other cheek elements, the postorbital is likely a small quadrangular element that extends from the top of the orbital margin adjacent to the postfrontal, ventrally towards the jugal, and contacts the squamosal posteriorly. It appears the small width of the postorbital is responsible for the short appearance of the postorbital region. The squamosal forms the majority of the lateral cheek and is represented by a large plate-like element on the right side of the cranium, as well as a displaced left squamosal that is present near the left forelimb. The squamosal is taller than it is wide, and indicates a relatively high cheek region. The contacts with most other cheek and dorsal skull elements are not clear. On the right side, a relatively straight, ventral contact with the quadratojugal can be observed. The quadratojugal morphology itself is not well-known due to the posteriormost area of the element being damaged on the original fossil. The right quadratojugal is preserved in medial aspect like the rest of the cheek. It is roughly triangular in shape with the wider end being posteriorly located. Posteriorly, the quadratojugal is partially overlapped by the quadrate. The right quadrate is a long, cylindrical element. It is incomplete anteriorly and lacks the process that would meet the pterygoid. Ventrally the articular surface for the mandible is well preserved and shows a median depression between two condyles.

Because the skull is preserved in ventral aspect, the palate is well represented though disarticulated. The preserved elements of the palate include both vomers, the right ectopterygoid, the right pterygoid and the right palatine. All of these elements are covered in a shagreen of small denticles. The vomer is represented anteriorly in the skull and is a quadrangular element with few distinctive qualities. Both left and right vomers appear to be tapered anteriorly and create a space that likely accommodated the internal median processes of the premaxillae. The contacts with other elements can only be speculated upon, but it is likely that the vomer met the palatine posterolaterally and the pterygoid medially. A single partial right palatine is present as a wide quadrangular element that

bears an emargination on the anterolateral surface where the internal naris would reside. The right ectopterygoid is represented by an extremely thin, denticulate element. It is likely the element was slightly wider than this in life based on the apparently broken margins of the bone. The right pterygoid is well represented and lacks only the quadrate ramus. A prominent anterior ramus is seen in ventromedial aspect and bears a shagreen of denticles. It is possible that the anterior ramus of the pterygoid is slightly raised to a small boss, similar to that of other basal eureptiles, but this is difficult to determine given the lack of association with the other palatal elements. The most striking feature of the pterygoid is the well-developed transverse flange, which occurs at an approximately 90-degree angle to the midline margin of the anterior ramus. The teeth on the transverse flange are not well preserved on any peel, but a field of slightly larger teeth can be observed on the resin cast and latex peels.

The occiput is represented by a few elements, including the supraoccipital, a partial basioccipital and at least one exoccipital. The supraoccipital is a large plate-like element that is roughly butterfly shaped. The middle of the supraoccipital is slightly raised to form a small, sagitally oriented ridge. Underlying the supraoccipital is an amorphous element identified as the basioccipital by Carroll & Baird [1]. Little can be said about this element other than it is slightly concave in dorsal aspect. At least one exoccipital, probably the left, is present and disarticulated near the atlas and axis. It is an elongated, cylindrical element. A medially located depression on the exoccipital likely represents the location of the jugular foramen. An element identified by Carroll & Baird [1] as the atlas pleurocentrum may in fact be the other exoccipital; however, it is not preserved well enough to be confidently identified.

Lastly, there is a series of seven to nine scleral ossicles. Scleral ossicles are commonly preserved in Mazon Creek tetrapods. Those observed in temnospondyls from Mazon Creek are often small, quadrangular elements numbering around 22–24. Those observed in YPM 796 differ in being large, rectangular ossicles that more closely resemble those found in other amniotes, such as the captorhinid *Reiszorhinus* [16].

## 4.2. Mandible of *Cephalerpeton*

Preservation of the mandible of YPM 796 includes the entire right jaw in medial perspective and the left dentary in lateral aspect (figure 2). The mandible, in general, displays a very gracile morphology. This is in contrast to the more robust elements found in the specimens assigned here to *Carbonodraco lundi* gen et sp. nov. The lateral surface of the dentary shows a pitted and slightly rugose ornamentation characteristic of captorinids.

The left and right dentaries are thinly built with an unexpanded symphysis that tapers gradually anteriorly. On the right dentary, preserved in lingual aspect, the symphysis appears even thinner and is excavated towards the centre. There may be a splenial attached to the right dentary but no sutures can be discerned. Neither dentary preserves their coronoid processes. The right surangular, angular and articular are preserved in medial aspect. These postdentary bones make up over one third of the length of the lower jaw. The surangular is the best represented of the postdentary bones. It is an irregular, oval-shaped element in lateral aspect. The angular is represented by an elongated element, and the articular is a small, likely oval-shaped ossification at the posteriormost end of the jaw. Because the articular is preserved in medial aspect, the extent and shape of the lateral surface is unknown, this includes whether or not there was a well developed retroarticular process.

The dentition on the dentaries differs slightly from that of the upper jaws. The left dentary has 18 teeth in place with spaces for a few more teeth. Carroll & Baird [1] estimated 24 teeth, with which we agree. The dentition on the lower jaws consists of sharply pointed conical teeth that are recurved apically. At the anteriormost end of the dentary, the dentition is slightly enlarged and anteriorly directed similar to that of most 'protorothyridid' reptiles. Overall the dentition on the dentaries appears less wide and tall than that on either maxilla.

## 4.3. Postcranial anatomy of *Cephalerpeton*

Our interpretation of the postcranial anatomy is largely consistent with the detailed descriptions provided by Carroll & Baird [1]. Here we briefly overview the anatomy and provide a few updated comparisons. The postcranial skeleton consists mostly of the presacral vertebral column (including the atlas-axis vertebrae), dorsal ribs, ventral gastralia, pectoral girdle elements and forelimbs. The vertebral count proposed by Gregory [7] of 25 to 26 vertebrae was identified by Carroll & Baird [1] as erroneously including elements of the occiput into the cervical vertebral series. Whereas 23 presacral vertebrae can be identified, we agree with Carroll & Baird [1] that a 28 total presacral vertebrae count is plausible.

The proatlas is represented in *Cephalerpeton* by a small round ossification in between the occipital elements and the atlas vertebra. Although the atlantal vertebral components cannot be confidently identified, a number of elements were identified by Carroll & Baird [1], including the atlantal arch and pleurocentrum (here alternatively considered an exoccipital). We also note that unidentified elements (unlabelled in figure 2) adjacent to the right exoccipital may also represent elements of the atlas, such as the atlas intercentrum. The small cylindrical axial pleurocentrum and large somewhat fan-like axial arch are present. It appears as though the arches are fused to the pleurocentra throughout the vertebral column due to the tight association of the two. The observed line running between the two may simply be a crack. The neural arch morphology throughout the vertebral column remains consistent, with strongly overlapping and well-developed prezygapophysis and postzygapophysis. The neural arches also bear a small anterior excavation that is depressed into the base of the neural spine. The neural spines are only fully preserved on approximately the eight anteriormost vertebrae. There they can be seen to have somewhat rounded margins, possibly indicating they were weakly ossified. Their height is comparable to early reptiles such as *Anthracodromeus*, but also some varanopids and other early pelycosaurian-grade tetrapods. A small transverse process can also be observed on some of the vertebrae. The pleurocentra are formed by large elongated cylinders that are concave on the lateral surfaces and that do not bear any noticeable keels or ridges. Ventrally, wedged between each pleurocentrum is a small, likely crescent-shaped, intercentrum. The dorsal ribs are well-represented along the postcranial skeleton. All ribs are single headed. The cervical ribs have a short but thick bar-like morphology and develop slight recurvature moving posteriorly down the series. The dorsal ribs posterior to the pectoral girdle are the longest at approximately three times the vertebral length. These ribs are also thin and highly recurved. Towards the posterior end of the preserved column the ribs shorten to a thin bar-like morphology and indicate the approach of the pelvic region.

The pectoral girdle is represented by the cleithrum, clavicle, scapula and anterior coracoid (procoracoid). Overall the pectoral girdle is lightly built like that of early basal eureptiles, such as *Anthracodromeus*, *Paleothyris*, *Hylonomus*, *Brouffia* and early diapsids [1,12,17]. The left cleithrum is teardrop shaped and slightly concave indicating it is the interior surface. The clavicle is roughly horn shaped and widens medially towards where the head of the interclavicle would have been. The left clavicle appears to have a somewhat short lateral process, but this bone may be incomplete. The clavicular head is ornamented with linear grooves. Although a piece of the right clavicle was identified by Carroll & Baird [1] next to the left humerus, we regard this element as unidentifiable. A partial right scapula is also preserved and reveals its exterior surface. The anterior coracoid is ovoid in outline and slightly overlapped by the scapula.

The right and left forelimbs are well preserved with only the phalanges and most of the carpal elements lacking. Overall the long morphology of the forelimb elements closely resembles the forelimbs of early diapsids, such as *Spinoequalis* [11], *Petrolacosaurus* [4], and *Araeoscelis* [13], but differs from the squat and robust limb morphology or *Thuringothyris*, captorhinids, and recumbirostrans. Both humeri are preserved as long rod-like shafts with a small proximal end with a moderately developed head. The distal end is greatly expanded. Both proximal and distal ends appear to be rotated at about 90 degrees to one another. The left distal humerus is rotated to reveal its articular surface that includes a shallow capitulum, and weakly developed entepicondyle. As pointed out by Carroll & Baird [1], no ridges or supinator process can be found on the humerus. This may indicate immaturity in the animal.

The ulna and radius are represented in both forelimbs as long rod-like elements that are approximately equivalent in length and only slightly shorter than the humerus. This is shared with *Spinoequalis*, *Petrolacosaurus* and *Araeoscelis* [4,11,13]. The proximal and distal ends of these elements appear both small in width and poorly developed. The ulna has no olecranon process, unlike that of captorhinids such as *Ophisthodontosaurus* [18]. Five metacarpals are preserved on the right manus and two on the left manus. They are also long and rod-like, the greatest of which is half the length of the zeugopodial elements, likely indicating the manus was long, similar to that of *Anthracodromeus*. The right manus also shows two to three overlapping distal carpals that are roughly cuboid in shape.

One of the remarkable features of YPM 796 is the presence of a suite of ventral gastralia, as well as soft-tissue, integumentary impressions hugging the forelimbs. The gastralia are thin, elongate rods that are canted anteriorly to meet at the midline to form a chevron. They are of the 'reptilian' morphology in that they are thin and unornamented (no concentric growth lines), and thus are similar to those found in *Hylonomus*, *Anthracodromeus*, as well as in varanopids and some recumbirostrans.

# 5. Systematic palaeontology

Tetrapoda Jaekel, 1909
 Amniota Haeckel, 1866
 Parareptilia Olson, 1947
 Acleistorhinidae Daly, 1969
 *Carbonodraco* gen. nov.

**Etymology:** Generic name derived from the Latin words '*Carbo*' (coal), and '*Draco*' (serpent). Specific name is in honour of Dr Richard Lund, who found the holotype.

**Diagnosis:** As for the type and only species.
 *Carbonodraco lundi* sp. nov.
 =*Cephalerpeton* cf. *C. ventriarmatum* Reisz and Baird 1983

**Holotype:** CM 23055, a disarticulated skull that includes maxillae, left premaxilla, right lacrimal, left prefrontal, left parietal, left frontal, dentaries and, splenials, left surangular, and vomers; collected by Richard Lund, 1972.

**Referred Material:** NHMUK R. 2667, right mandible in lingual perspective; probably collected by John S. Newberry, circa 1870, later given to James W. Davis ('Davies' of Reisz & Baird [8]), and eventually purchased by the British Museum [Natural History] in 1895. CM 81536, a pair of dentaries preserved in lingual perspective; collected by Scott McKenzie, 2004.

**Locality and Horizon:** Coal mine operated originally by the Ohio Diamond Coal Company at Linton, Saline Township, Jefferson County, Ohio, USA (see [19] for details). Local cannel coal immediately below the Upper Freeport coal, Allegheny Group, Middle Pennsylvanian (Moscovian).

**Diagnosis:** An acleistorhinid parareptile diagnosed by the following unique combination of characters: vomers covered in a shagreen of denticles; parietals wide; pineal foramen anteriorly located. Additional characters shared with acleistorhinids: two enlarged anterior teeth on maxillae and anteriormost premaxillary tooth enlarged to the height of enlarged maxillary teeth (shared with *Colobomycter*); lacrimal short and excluded from external naris (shared with *Colobomycter* and *Acleistorhinus*); high facial lamina of the maxilla and pitted ornamentation shared with all acleistorhinids.

# 6. Description

A left premaxilla is preserved in the holotype specimen. It appears to have a short lateral process and a high dorsal ascending process. The lateral surface is ornamented with tiny pits that probably are foramina. The premaxilla has spaces for approximately three teeth, two of which are in place, and one of which is significantly enlarged. This large tooth is the anterior most, and its length equals that of the most enlarged maxillary teeth. All premaxillary teeth are gently recurved towards the apex. Along the base of the enlarged premaxillary tooth there can be seen large grooves likely indicating the presence of plicidentine. Overall the structure and size of the tooth is most similar to that of the parareptile *Colobomycter vaughni*, whereas the tooth in *Colobomycter pholeter* is even larger [20–22].

CM 23055 preserves the right and left maxilla in lateral view (figures 3 and 4). The ornamentation consists of distinct, large pitting similar to that of other known parareptilian taxa, including acleistorhinids such as *Colobomycter* and *Delorhynchus*. The large pitting on the facial lamina of the maxillae is highly comparable to that of *Colobomycter pholeter* and *Colobomycter vaughni* [20,23]. The facial lamina of the maxilla is tall and narrows slightly dorsally. The subnarial process of the maxilla bears an anterolateral foramen adjacent to the naris similar to the maxilla of *Acleistorhinus*, *Colobomycter* and other parareptiles. The posterior process tapers significantly away from the facial lamina. The conical and sharply pointed tooth crowns display substantial heterodonty in terms of size along the tooth row. Two enlarged teeth of similar size, at positions four and five, are significantly larger than the rest. Many of the teeth clearly show linear grooves that begin at the tooth base and end midway towards the crown.

The lacrimal is represented only on the left side and is preserved in medial view in CM 23055. It forms a portion of the anterior orbital margin and is comparatively shorter than that of basal reptiles, such as 'protorothyridids' and captorhinids, and also many recumbirostran taxa. Although Reisz & Baird [8] figured the lacrimal as being incomplete anteriorly, we find this unlikely because an anterior margin is present on both the latex peel and original specimen. The short morphology of the lacrimal

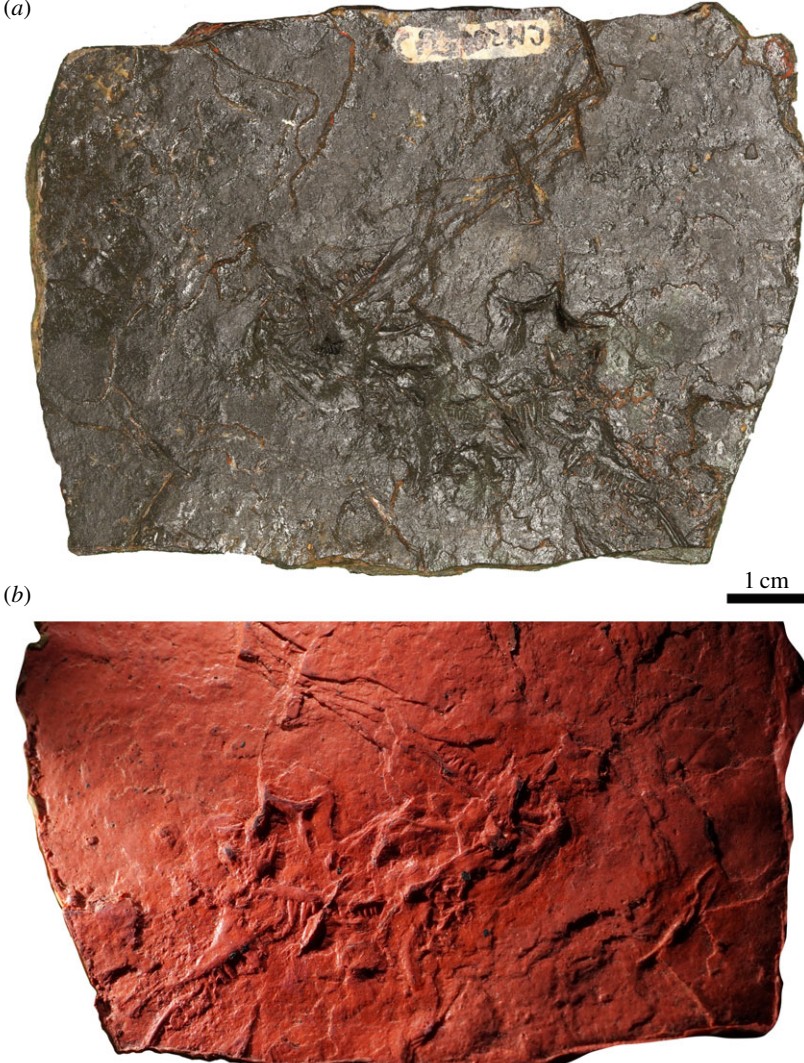

**Figure 3.** *Carbonodraco lundi* gen. et sp. nov., (*a*) the holotype specimen (CM 23055) (Amy C. Henrici photo) and (*b*) latex peel of the holotype specimen.

resembles that of some acleistorhinid parareptiles (*Acleistorhinus*, *Colobomycter* and some specimens of *Delorhynchus*), where the lacrimal is excluded from the naris and partially overlapped by the high facial lamina of the maxilla [20]. A long groove that accepts the ventral process of the prefrontal is present on the posterodorsal edge of the medial surface of the lacrimal. The posteroventral process of the lacrimal is short, dissimilar to that of *Cephalerpeton* and other basal eureptiles, but close morphologically to that of *Colobomycter* and *Acleistorhinus*.

A single left prefrontal is preserved in CM 23055, also in medial aspect. This element is roughly Y-shaped with all of the processes approximately the same length and width. The ventral process can be elegantly matched with the posterodorsal recess on the lacrimal. This ventral process of the prefrontal, in conjunction with the posterior process of the prefrontal, form the anterodorsal margin of the orbit.

A right nasal is tentatively identified in CM 23055, disarticulated and now located between the vomers. The posterior margin of the nasal is obscured by the right vomer, but its anterior margin is visible and reveals what appears to be the dorsal surface. There is an anterolateral flange that extends to what may be the anterior margin of the external naris. Anteromedially, there is recess that may have housed the ascending dorsal process of the premaxilla.

The left frontal is preserved in ventral aspect, and the right frontal may also be present, overlapped by the anterior end of the left. The bone is long and rectangular with a small median lappet that likely formed a contribution to the dorsal margin of the orbit. Overall, the frontal is also quite wide, and both frontals together would have formed a wide interorbital region. This is another feature that is drastically different from the condition seen in *Cephalerpeton* but similar that of acleistorhinid parareptiles.

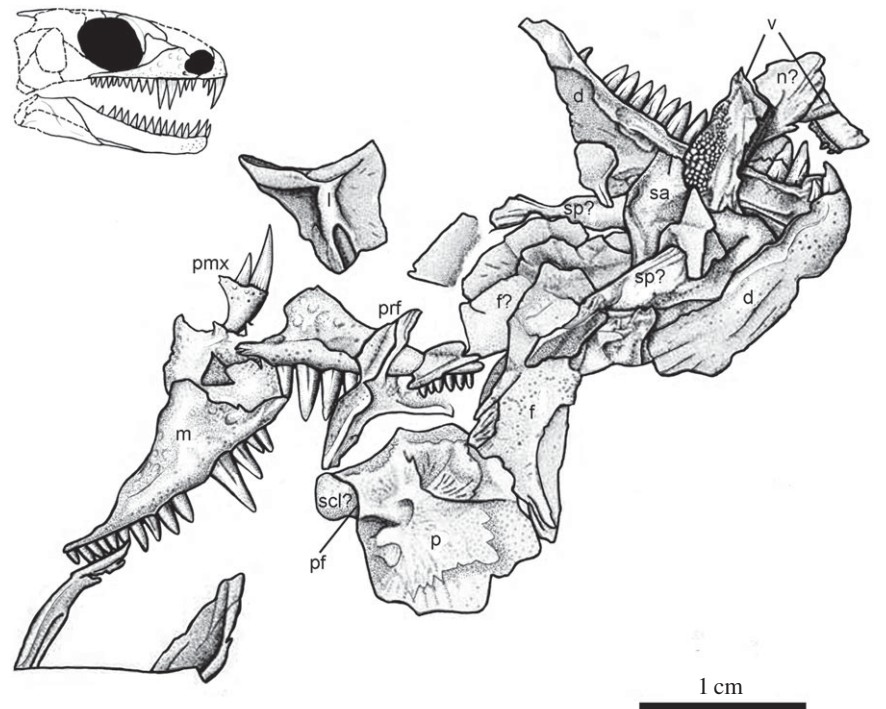

**Figure 4.** Illustration of the cranial anatomy of the holotype of *Carbonodraco lundi* gen. et sp nov. (CM 23055), and a reconstruction.

The parietals are represented only by the left element that is preserved in ventral aspect. It is basically square in outline and is very wide in comparison to that of most 'protorothyridids', early eureptiles and even most parareptiles. The straight anterior margin of the parietal indicates that the suture with the frontal was likely simple and not strongly interdigitated. The posteromedial margin of the parietal is emarginated slightly to create, together with its antimere, a recess for the postparietal. The parietal-parietal contact is straight with the exception of a small emargination for the anteriorly located pineal foramen. This anterior location of the pineal foramen is unique to *Carbonodraco* in comparison to other acleistorhinids. The ventral surface is ornamented with striae and bears a slightly raised medial surface that extends from the posterolateral edge.

The only palatal elements identifiable in CM 23055 are the vomers. The right vomer is preserved in ventromedial aspect, while the left vomer is only visible in lateral aspect. They both show a high median lamina. The right vomer is better exposed and shows that the vomers are roughly triangular in ventral aspect. The right vomer narrows anteriorly towards the internal median premaxillary contact and widens posteriorly towards both the palatine and pterygoid. The vomer is covered in a shagreen of denticles. This is unique for a parareptile, where the vomers often bear a continuation of teeth from a denticulate boss on the pterygoid. The vomerine dentition of *Carbonodraco* may be primitive in this regard.

The mandible of *Carbonodraco lundi* gen. et sp. nov. is represented by both dentaries, two tentatively identified splenials, and a left surangular (figures 4 and 5). The dentary is by far the best represented mandibular element and is preserved on the holotype specimen (CM 23055), as well as the two referred specimens (NHMUK R. 2667 and CM 81536). The dentary is comparatively robust, particularly in the area of the slightly upturned symphysis. The lateral surface has some gentle rugosity and small foramina anteriorly. Medially, there are fine ridges preserved that may indicate the location of at least one coronoid ventral to a relatively low coronoid process.

The tooth-count estimates vary between specimens but are within the range of variation seen in extant reptiles [24]. Whereas the dentaries of CM 23055 have 16 to 17 tooth positions, which agrees with 16 positions in NHMUK R. 2667, CM 81536 has places for 19 teeth. The teeth of CM 81536 bear only a weak degree of heterodonty, with the anteriormost dentary teeth being slightly larger than the rest. Unlike *Cephalerpeton*, the dentary teeth of *Carbonodraco lundi* gen et sp. nov. are approximately the same size, if not larger, than the opposing teeth on the maxillae and are of the same morphology in that they are not recurved. CM 81536 preserves details of the dentition better than any other specimen of *Carbonodraco lundi* gen et sp. nov. The morphology of each individual tooth is conical and tapered more abruptly in the apical portion of the tooth crown. Each tooth crown bears distinct enamel fluting consisting of very fine parallel grooves. The tooth bases and midsections also show

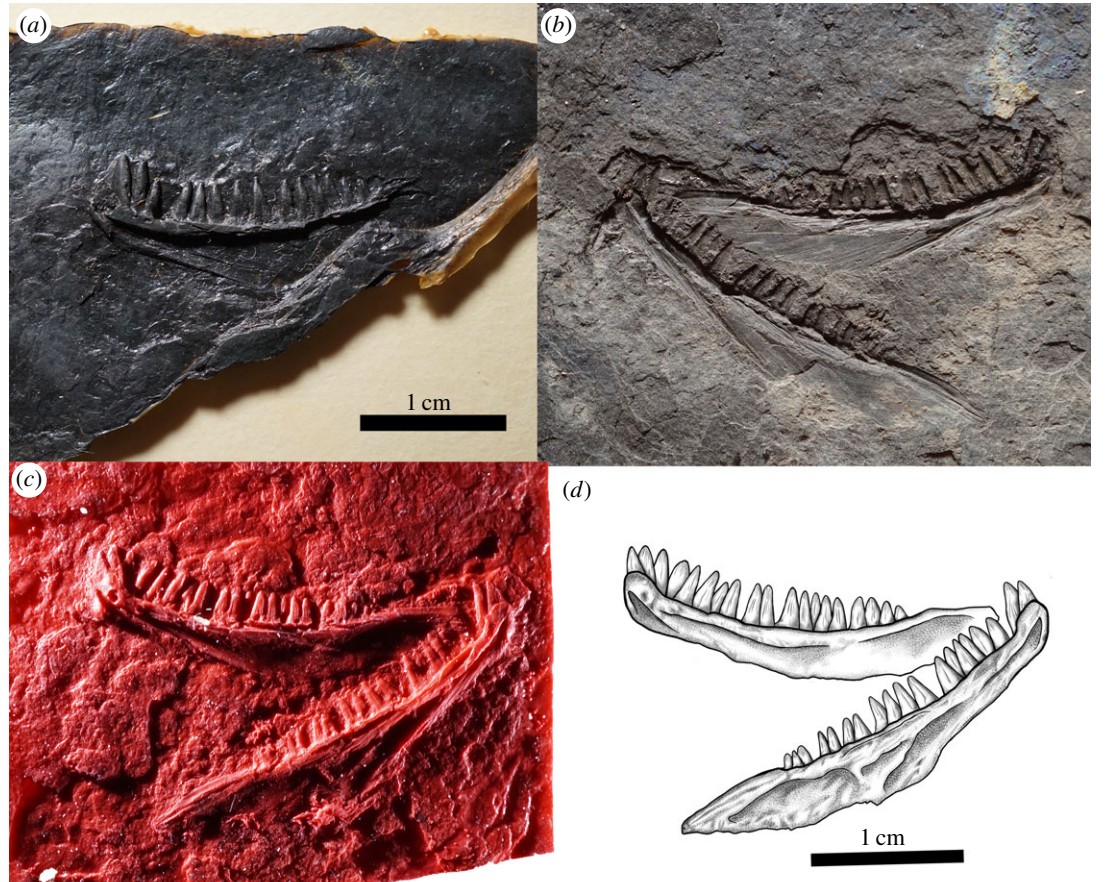

**Figure 5.** Referred specimens of *Carbonodraco lundi* gen. et sp. nov. (*a*) Latex peel of NHMUK R. 2667 showing a right jaw in lingual perspective (Kathy Bossy peel). (*b*–*d*) CM 81536. (*b*) Original cannel coal specimen, (*c*) latex peel, and (*d*) drawing, showing a pair of dentaries preserved in lingual perspective.

large grooves, often three or more. These larger grooves are interpreted as plicidentine, similar to that found in parareptiles such as *Colobomycter*, as well as a variety of other Palaeozoic tetrapods.

The splenial is a flat, long element with a posterior lappet that may have cupped the posterior end of the dentary (CM 23055, figure 4). The left splenial is located adjacent to the left dentary and appears to be preserved showing its exterior surface, whereas the right splenial appears to be showing its lingual surface. The left surangular is preserved partially and overlapped by the right splenial. It has a moderately developed crest, which is confluent with the coronoid process of the dentary, and a slightly concave lateral surface.

# 7. Systematic palaeontology

Tetrapoda Jaekel, 1909
    Amniota Haeckel, 1866
    Reptilia indet.

**Material:** CM 41714, a partial left jaw, including a dentary (broken into two pieces), splenial, angular and surangular, preserved in lateral view, as well as rib fragments and a partial centrum; collected by Carl F. Wellstead, 1983.

**Locality and Horizon:** Coal mine operated originally by the Ohio Diamond Coal Company at Linton, Saline Township, Jefferson County, Ohio, USA (see [19] for details). Local cannel coal immediately below the Upper Freeport coal, Allegheny Group, Middle Pennsylvanian (Moscovian).

**Comment:** The general mandibular morphology indicates a relatively long-jawed reptile unlike *Carbonodraco*. There is no significant anatomical overlap between this specimen and the solitary Linton

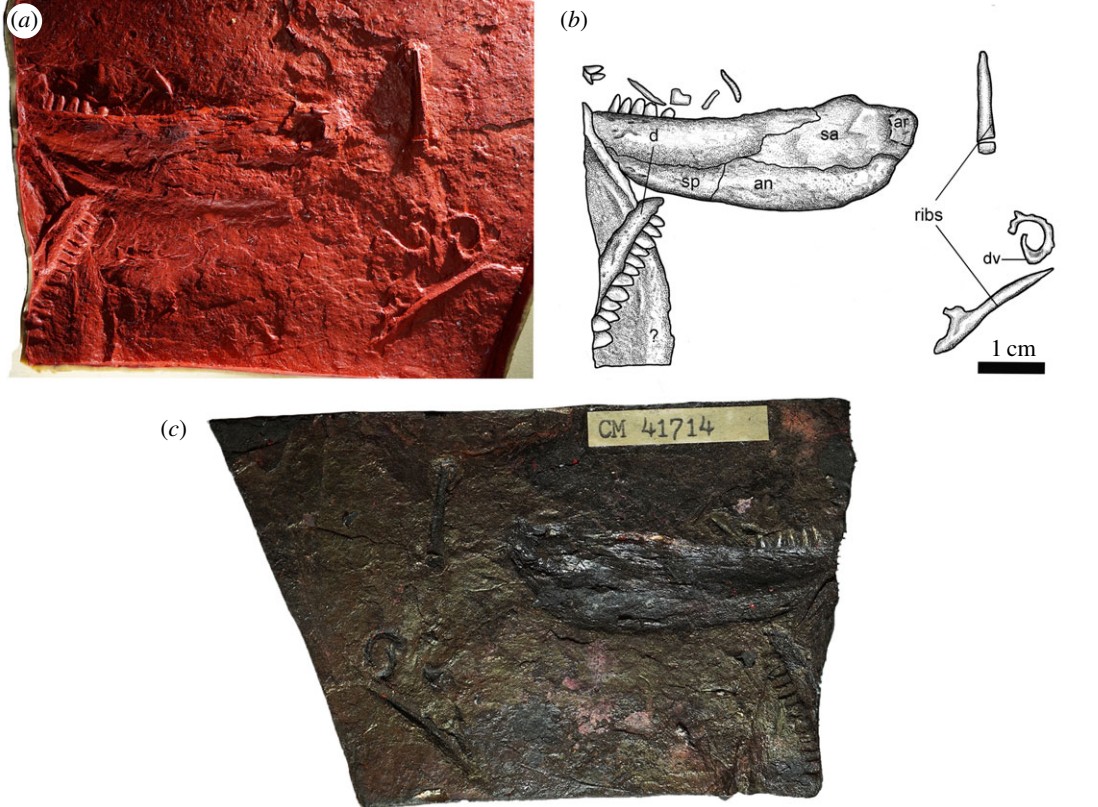

**Figure 6.** Indeterminate reptilian amniote from Linton, Ohio (CM 41714). (*a*) Latex peel, (*b*) drawing, and (*c*) actual specimen.

specimen that represents *Anthracodromeus* (see [1]). Thus, we recognize CM 41714 as a distinct, but indeterminate, reptile taxon in the Linton assemblage.

# 8. Description

CM 41714 includes a left jaw and associated postcranial fragments that bear some resemblance to early reptiles (figure 6). The preserved mandible includes a dentary (the anteriormost portion of which is broken and displaced), splenial, angular, surangular and articular, all preserved in lateral view. Postcranial fragments include a few rib fragments and a partial centrum. Additionally, there is a large unidentifiable bone located on the edge of the block beneath the anterior dentary fragment.

The anterior portion of the dentary shows a rapidly narrowing morphology that ends in a small unexpanded symphysis; this is unlike the dentary of *Carbonodraco*. The posterior portion of the dentary is deep dorsoventrally and has a strongly convex ventral margin. The lateral surface shows a slightly rugose ornamentation, with small foramina dispersed throughout, similar to the ornamentation seen in some captorhinid eureptiles. In total, seventeen teeth are preserved, with spaces for more at the posterior end. The teeth are homodont, with wide, bulbous bases. The crowns are worn, yet they retain a somewhat pointed apex. Although some captorhinids and recumbirostrans show a similar tooth morphology, the teeth of CM 41714 are insufficient for detailed comparisons. Unlike *Carbonodraco* or *Cephalerpeton*, no evidence of enamel fluting or plicidentine grooves can be found on the teeth of CM 41714.

The splenial is elongate and buttresses the ventral surface of the dentary. Its ventral margin continues the convex shape of the entire jaw. The splenial also bears rugose ornamentation and pitting similar to the dentary. Posteriorly, the splenial contacts the angular in a simple straight suture. Together, the surangular and angular contribute to about a third of the mandible length. The surangular is a large irregularly-shaped element that bears a low coronoid process about midway along its dorsal margin. At the posterodorsal edge of the surangular, a small quadrangular ossification probably represents an articular; the vague expression of this element suggests that it may have been poorly ossified. The angular occupies the posteroventral region of the lower jaw. Its ventral margin is strongly curved, especially at its posterior terminus.

# 9. Discussion

## 9.1. *Cephalerpeton ventriarmatum*, an early eureptile from Mazon Creek

Here we conducted the first restudy of the anatomy and systematic considerations of early reptiles from the Pennsylvanian-aged sites of Mazon Creek, Illinois, and Linton, Ohio, since the 1980s. Our anatomical analysis supports the placement of *Cephalerpeton ventriarmatum* within Eureptilia but found characteristics in common with both Carboniferous basal eureptiles and more derived diapsid eureptiles, such as the araeoscelids. Taking into account that Müller & Reisz [5] recovered *Cephalerpeton* as a member of the sister clade (together with *Anthracodromeus* and *Protorothyris*) to Diapsida, and that the oldest araeoscelids occur in the Late Carboniferous (Kasimovian), it is not unreasonable to consider *Cephalerpeton* as more closely related to diapsids than previously thought. This phylogenetic position, and the question of whether or not *Cephalerpeton* (and by proxy, *Anthracodromeus* and *Protorothyris*) should be included within Diapsida, remains a possibility that needs to be tested with a comprehensive phylogenetic analysis. We did not perform such an analysis here because we believe that it would be premature in the absence of revisions of other relevant early reptiles. At present, our understanding of the anatomy of most Pennsylvanian-aged reptiles (i.e. most members of Protorothyrididae) is somewhat dated, and was generated when a much smaller comparative database was available. Additionally, several recent studies have proposed major shifts in the amniote phylogenetic tree, including the inclusion of recumbirostrans as early reptiles [25] and varanopids as early diapsids [26]. A comprehensive reevaluation of early amniote phylogeny, synthesizing and testing these new hypotheses, is the appropriate first step towards a better understanding of enigmatic Carboniferous forms such as *Cephalerpeton*.

Our restudy of the anatomy of *Cephalerpeton ventriarmatum* provides a revised diagnosis for the genus and recognizes that the genus is restricted to Mazon Creek. Previously referred material from Linton, Ohio, is identified here as the new parareptile *Carbonodraco lundi*. Although we restrict *Cephalerpeton* remains at this time to only occurring in the Mazon Creek assemblage, without the complete revision of reptile remains from Linton, Ohio, including the closely related *Anthracodromeus*, as well as reptile remains from other Carboniferous sites, there is no way to be certain about the level of endemism occurring in closely aged Carboniferous amniote communities.

*Cephalerpeton ventriarmatum* represents the oldest known reptile aside from *Hylonomus* from Joggins, Nova Scotia. Unlike the upright tree stump assemblages of Nova Scotia, *Cephalerpeton* occurs in a siderite nodule at Mazon Creek, which is believed to represent an estuarine setting [27]. The Mazon Creek assemblage is dominated by aquatic invertebrates, insects and fish [28]. Tetrapods found in Mazon Creek nodules are believed to have been washed in from the adjacent near-shore environment [27]. In contrast to this traditional scenario, recent studies of tetrapods from this locality have identified a number of terrestrial tetrapods in the assemblage [29,30]. Interestingly, these recumbirostrans display several adaptations to fossoriality, suggesting the terrestrial component of Mazon Creek tetrapods has been understated. Additionally, recumbirostrans were recently recovered as a group of reptiles [25] revealing the estuarine palaeoenvironment of the Mazon Creek lagerstätte was likely home to a significant diversity of reptile and reptile-like terrestrial taxa.

Traditional hypotheses of amniote origins have described the establishment of dry, 'upland' ecosystems as a possible driver of early amniote diversification. Testing of this idea has been lacking until the recent attempts of Pardo *et al.* [31]. Records of terrestrial amniotes (including synapsids, eureptiles and parareptiles) at Mazon Creek and Linton—clearly lower delta plain and alluviated upper delta plain settings, respectively—also suggests a more complicated palaeoenvironmental and ecological scenario for early amniote diversification.

There are some important early ecological adaptations revealed by the current reanalysis of the anatomy of *Cephalerpeton*. The most obvious of these is the dental configuration, which includes large, wide-based conical teeth and multiple enlarged teeth irregularly spaced along the maxillary tooth row with smaller recurved teeth in the dentary tooth row. Carroll & Baird [1] and Reisz & Baird [8] described the maxillary dentition of *Cephalerpeton* as adequate for the processing of hard-shelled arthropod material, which represent a widely available food source at Mazon Creek and most Carboniferous coal measure localities [32]. We agree that the teeth were likely used for a form of durophagous carnivory/insectivory. The wide, conical teeth with multiple peaks along the tooth row would have done well piercing the chitinous exoskeletons of small insects, as the lower dentition held prey in place (hence the recurvature) (see [33] for interpretation of tooth types in reptiles). The dentition of *Cephalerpeton* seems to represents an alternative approach to insectivorous durophagy

contrasting with the very wide, teardrop-shaped dentition present in durophagous gymnarthrids, pantylids, the captorhinid *Opisthodontosaurus*, and durophagous modern squamates [18,33,34]. The ogival tooth morphology seen in other captorhinids is yet another example of durophagous dentition among early reptiles [35]. This 'captorhinid' type dentition could be convergently shared with CM 41714, the indeterminate reptile from Linton, Ohio. Additionally, similar maxillary teeth are seen in *Cephalerpeton* and *Carbonodraco*, as well as a number of parareptiles [32], which suggests that this durophagus tooth morphology may also be prone to convergence.

Another interesting aspect of *Cephalerpeton* is the length of the forelimbs, which are proportionally longer than those present in most 'protorothyridids' with the exception of the closely related *Anthracodromeus*. As described above, the length of the forelimbs more closely resembles the proportions of those of later occurring diapsids (e.g. *Petrolacosaurus*). Carroll & Baird [1] recognized similarities in limb proportions and morphology of the manus and pes of the Linton taxon *Anthracodromeus* to those of extant arboreal reptiles. Given the extensive lycopsid forests present in the Carboniferous, the idea that *Anthracodromeus* was arboreal is certainly plausible. As such, it is also plausible that *Cephalerpeton* was also adapted for some form of arboreal or at least scansorial lifestyle.

## 9.2. *Carbonodraco lundi* as the oldest parareptile

The identification of CM 23055 as a distinct taxon from *Cephalerpeton* increases the taxonomic diversity of amniotes at Linton, which now includes representatives of Synapsida, Eureptilia and Parareptilia. Our interpretation of the taxonomic affinity of *Carbonodraco* further makes it the oldest known member of Parareptilia, the early diverging sister clade to Eureptilia. Previously, the oldest known parareptile was *Erpetonyx arsenaultorum* [36], from the uppermost Carboniferous (Gzhelian) Egmont Bay Formation of Prince Edward Island, Canada. *Erpetonyx* is a generalized parareptile and was placed in the relatively early-branching clade Bolosauria. Modesto *et al.* [36] also established that parareptiles began their evolutionary radiation before the end of the Carboniferous Period. As a result, their time-calibrated phylogeny of parareptiles revealed long ghost lineages for basal parareptilian clades (e.g. Mesosauridae and Millerosauria) since these taxa appear in the fossil record in the Permian. The evolutionary framework presented in this time-calibrated phylogeny additionally shows an Early Permian diversification of the clade Ankyramorpha (i.e. lanthanosuchids, nyctiphuretids and procolophonids). Recent studies by MacDdougall *et al.* [37] on parareptile species richness and diversity through time also support this framework. Thus, our identification of *Carbonodraco* as an acleistorhinid parareptile is of high importance to the interpretation of the timing and diversification of the previously identified Permian-aged adaptive radiation of not only the Ankyramopha, but the entirety of the parareptile clades to as early as the Moscovian. The occurrence of *Carbonodraco lundi* gen et sp. nov. in the Carboniferous aligns the first appearance of parareptiles close to that of eureptiles represented by *Hylonomus* in the Bashkirian, though many long ghost lineages for parareptiles still remain.

It now seems likely that the previously identified timing of parareptile evolution is an artefact derived from a sampling bias of Permian-aged localities, such as Richards Spur. This revelation highlights the importance of research on new and existing tetrapod fossils from earlier, Carboniferous-aged localities. Furthermore, phylogenetic analyses of various reptilian groups need to be less exclusive and take into account wider taxonomic sampling from groups including early eureptiles [5], early diapsids [26] and recumbirostrans [25]. Inclusions of these groups and revised anatomical analyses have the potential to alter currently accepted evolutionary relationships of Reptilia.

Ethics. No ethics assessment was required prior to the completion of this research because this study relied entirely on museum collections. Similarly, collecting permits were not required because no field collections were made.

Data accessibility. All data, which include photographs and text description of fossils, are included in the paper.

Authors' contributions. A.M. designed the study; A.M. E.R.M. and E.J.M. collected and analysed the data; A.M. and H.C.M. wrote the paper.

Competing interests. We declare we have no competing interests.

Funding. We received no funding for this study.

Acknowledgements. We thank Amy C. Henrici and David S Berman for facilitating work at the Carnegie Museum of Natural History. Diane Scott and Robert R. Reisz (University of Toronto) allowed access to parareptile material from Oklahoma and Texas localities under their care. Bryan M. Gee, Sean P. Modesto and Jason D. Pardo provided stimulating discussions. The late John Bolt was a much valued colleague who contributed directly to this work. Likewise, the late Donald Baird prepared much of the material reported in this study; we thank Robert W. Hook for access to Baird's original research materials. We also thank three anonymous reviewers for their helpful

comments. Lastly but perhaps foremost, we thank Richard Lund, Scott McKenzie and Carl F. Wellstead for their generous donations of specimens from Linton that form the basis of much of this research.

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
