## [Reviewer comments · Royal Society Open Science]

Review History

RSOS-191191.R0 (Original submission)

Review form: Reviewer 1

Is the manuscript scientifically sound in its present form?

Yes

Are the interpretations and conclusions justified by the results?

Yes

Is the language acceptable?

Yes

Do you have any ethical concerns with this paper?

No

Have you any concerns about statistical analyses in this paper?

No

Recommendation?

Accept with minor revision (please list in comments)

Comments to the Author(s)

This is a nice little paper that clarifies the amniote presence at the Carboniferous of the U.S.A. After minor revisions, it would be eminently suitable for publication in Royal Society Open Science. I think the paper would be greatly enhanced by an updated list(s) of the vertebrate/tetrapod fauna now recognized at the Mazon Creek and Linton sites.

I have the following minor comments/corrections (based on the pagination in the Word document):

Line 46: typo in 'protorthyridids' (here and elsewhere in text)

Line 53: typo in 'acliestorhinids' (here and elsewhere in text, e.g. line 442)

Line 64: the authors use the term 'sauropsids' whereas they use the term 'reptiles' in the title and elsewhere in their paper; it is not clear if the authors are using these terms interchangeably or if they refer to different groupings of amniotes; if the former, the authors should pick one term and use it consistently

Line 70: the locality name 'Nyraný' is missing diacritical marks [Nýřany]

Line 73: given my comments for line 64, I am not sure what the authors mean by 'stem-reptile family': do they mean that protorthyridids are stem reptiles sensu Robert Carroll? Or do they mean that they regard Reptilia to be a crown group and thus Sauropsida is the total group? Please clarify.

Line 118: typo in 'Acliestorhinus'

Line 129: there are two abbreviations for 'exoccipital' ('eo' is used in figure 2b)

Line 164: 'Macdougall' should be 'MacDougall' (here and elsewhere in text)

Line 178: one can estimate skull length, or measure the exact length of a skull, but not 'estimate' the exact skull length

Line 306: change 'cuboidal' to 'quadrangular'

Line 311: each gnathostome has a single mandible (there is no such thing as a right mandible in tetrapod biology, unless you are describing a teratology); see also lines 435 and 909

Line 314: delete one of the full stops following 'nov'. See also lines 547, 589

Line 322: change 'however, no sutures can be discerned' to 'but no sutures can be discerned'

Line 330: change 'which is likely accurate' to 'with which we agree'

Line 421: excuse my pedantry, but the Latin 'draco' is from ancient Greek 'drakon', which means 'serpent' and was used to refer to snakes of unusual size ('ophis' was used for small snakes). That is to say, the ancient Romans and Greeks did not associate their terms with (mythological) winged reptiles, as this is a later concept (the English 'dragon' is derived from 'draco'). I am not

suggesting a name change for the authors' new genus, but they should be clear on the etymology; authors may wish to check out the book "Dragon Myth and Serpent Cult in the Greek and Roman Worlds" by Daniel Ogden for background.

Line 436: typo in 'Riesz'

Line 449: change 'are not able to be identified at present' to 'are not identifiable at present'

Line 461: change 'recurvated' to 'recurved'

Line 657: a reference is needed here

Line 670: I have worked on extinct reptiles for many years, and I find the phrase 'more conventional reptilian bauplan' puzzling. Please elaborate on this concept.

Line 781: author name is 'deBraga'

Lines 790, 793, and 800: information missing from these references

Figure 2: label C missing from skull reconstruction

Review form: Reviewer 2 (Juan C. Cisneros)

Is the manuscript scientifically sound in its present form?

Yes

Are the interpretations and conclusions justified by the results?

Yes

Is the language acceptable?

Yes

Do you have any ethical concerns with this paper?

No

Have you any concerns about statistical analyses in this paper?

No

Recommendation?

Accept with minor revision (please list in comments)

Comments to the Author(s)

This is an interesting paper that extends the lineage of one important clade of Permian reptiles. The authors have nicely demonstrated that Carbonodraco is a parareptile.

I request the authors to use the term Reptilia, this name clearly has priority over the unnecessary "Sauropsida". See Modesto and Anderson (2004) for a phylogenetic definition of Reptilia.

I made some minor comments and corrected some typos (see Appendix A). The names *Acleistorhinus* and *acleistorhinids* are misspelled throughout the entire text. Please also check my comments to the illustrations in that file.

Review form: Reviewer 3 (Kirstin S. Brink)

Is the manuscript scientifically sound in its present form?

Yes

Are the interpretations and conclusions justified by the results?

Yes

Is the language acceptable?

Yes

Do you have any ethical concerns with this paper?

No

Have you any concerns about statistical analyses in this paper?

No

Recommendation?

Accept with minor revision (please list in comments)

Comments to the Author(s)

This paper re-describes some interesting specimens representing early eureptilians and parareptiles, which are important for understanding the origin and evolution of Reptilia. The descriptions are thorough, and my comments mainly refer to the terminology used to describe the teeth:

Line 99: “crafted by Donald Baird” sounds like he made it up. Maybe say “... we were able to study original latex peels taken/collected/made by DB and casts of...”

Line 156: “maxilla bears multiple canine-like peaks”. What does this mean? Are all the teeth caniniform in shape? Using the word ‘teeth’ instead of peaks would make much more sense here, if this statement is actually referring to teeth and not some peaks made of bone on the maxilla. If this statement is referring to tooth size, maybe say: Maxillary teeth irregular in size along toothrow/ maxillary tooth height inconsistent along toothrow/ or something like that. See comments below regarding the use of ‘canine’ and separating tooth position, tooth size, and tooth shape in descriptions.

Line 188: “There a thin and delicately built anatomy is seen...” Add a comma after “There”.

Line 195: The lacrimal

Line 199: The word “canines” in parentheses is not necessary here, since canine refers to a specific tooth type and tooth position in mammals. See comments below.

Line 215: Sides missing an S.

Line 320: confusing sentence “On the right, lingually preserved, dentary the symphysis...”

Line 444, 460, 462, 474: the use of ‘canine’ and ‘incisor’ is problematic, as these are mammalian terms referring to a specific location in the mouth and specific tooth shapes. Check out the discussion in Macdougall and Reisz 2014 on the use of the term ‘canine’ and ‘caniniform’ and switch your language accordingly (talk about size differences in term of tooth number, e.g., maxillary teeth 4 and 5 are twice the length of all other maxillary teeth). This will make scoring these animals in future phylogenetic analyses easier. Separate tooth size from tooth shape and tooth position.

MacDougall, M. J., and R. R. Reisz. 2014. The first record of a nyctiphruetid parareptile from the Early Permian of North America, with a discussion of parareptilian temporal fenestration. *Zoological Journal of the Linnean Society* 172:616-630.

Line 449: “are not able to be identified at present” can be “are not identifiable at present”

Line 461: “gently recurvated” should be gently recurved

Line 542: ‘...have places of 19 teeth...’ should be ‘for’ instead of ‘of’

Line 588: ‘... narrowing morphology that end with a...’ missing s

Line 650: Pardo et al. (2019) missing from reference list.

Lines 651-655 are very confusing and should be re-written for clarity, especially ‘Upland ecosystems are no more abundant in early amniotes than other ecosystems’

Line 657 (reference)

Line 673, 679 ‘multiple canine peaks in the maxillary tooth row’. See comments above. Separate tooth shape from tooth size and tooth position.

Line 678-680: Can you cite something that you would support your interpretation of this type of feeding?

Line 698: ‘had arboreal habits’ should be ‘was arboreal’

Figure 2: The peel doesn’t offer much information. Is it possible to make the drawing bigger in order to better see the described anatomy? This would make the description much easier to follow (especially the palate). The scleral ossicles should be included in the cranial reconstruction.

Figure 3: This figure doesn’t show anything informative. Maybe combine with figure 4, but maintain the detail and size of the interpretive line drawing? Consider putting photos of peels as supp info and just focus on the detailed drawings for the figures in the paper?

References: most species names missing italics

Decision letter (RSOS-191191.R0)

13-Aug-2019

Dear Mr Mann

On behalf of the Editors, I am pleased to inform you that your Manuscript RSOS-191191 entitled "Reappraisal of Carboniferous early reptiles from Mazon Creek and Linton reveals the oldest parareptile: *Carbondraco lundii* gen. et sp. nov." has been accepted for publication in Royal Society Open Science subject to minor revision in accordance with the referee suggestions. Please find the referees' comments at the end of this email.

The reviewers and handling editors have recommended publication, but also suggest some minor revisions to your manuscript. Therefore, I invite you to respond to the comments and revise your manuscript.

- Ethics statement

- Data accessibility

If you wish to submit your supporting data or code to Dryad (<http://datadryad.org/>), or modify your current submission to dryad, please use the following link:
<http://datadryad.org/submit?journalID=RSOS&manu=RSOS-191191>

- Competing interests

- Authors' contributions

- Acknowledgements

- Funding statement

Because the schedule for publication is very tight, it is a condition of publication that you submit the revised version of your manuscript before 22-Aug-2019. Please note that the revision deadline will expire at 00.00am on this date. If you do not think you will be able to meet this date please let me know immediately.

- 1) A text file of the manuscript (tex, txt, rtf, docx or doc), references, tables (including captions) and figure captions. Do not upload a PDF as your "Main Document";
- 2) A separate electronic file of each figure (EPS or print-quality PDF preferred (either format should be produced directly from original creation package), or original software format);
- 3) Included a 100 word media summary of your paper when requested at submission. Please ensure you have entered correct contact details (email, institution and telephone) in your user account;
- 4) Included the raw data to support the claims made in your paper. You can either include your data as electronic supplementary material or upload to a repository and include the relevant doi

within your manuscript. Make sure it is clear in your data accessibility statement how the data can be accessed;

5) All supplementary materials accompanying an accepted article will be treated as in their final form. Note that the Royal Society will neither edit nor typeset supplementary material and it will be hosted as provided. Please ensure that the supplementary material includes the paper details where possible (authors, article title, journal name).

on behalf of Dr Julia Brenda Desojo (Associate Editor) and Kevin Padian (Subject Editor)
openscience@royalsociety.org

Reviewer comments to Author:
Reviewer: 1

Comments to the Author(s)

This is a nice little paper that clarifies the amniote presence at the Carboniferous of the U.S.A. After minor revisions, it would be eminently suitable for publication in Royal Society Open Science. I think the paper would be greatly enhanced by an updated list(s) of the vertebrate/tetrapod fauna now recognized at the Mazon Creek and Linton sites.

I have the following minor comments/corrections (based on the pagination in the Word document):

Line 46: typo in 'protorothyridids' (here and elsewhere in text)

Line 53: typo in 'acliestorhinids' (here and elsewhere in text, e.g. line 442)

Line 64: the authors use the term 'sauropsids' whereas they use the term 'reptiles' in the title and elsewhere in their paper; it is not clear if the authors are using these terms interchangeably or if they refer to different groupings of amniotes; if the former, the authors should pick one term and use it consistently

Line 70: the locality name 'Nyraný' is missing diacritical marks [Nýřany]

Line 73: given my comments for line 64, I am not sure what the authors mean by 'stem-reptile family' : do they mean that protorothyridids are stem reptiles sensu Robert Carroll? Or do they mean that they regard Reptilia to be a crown group and thus Sauropsida is the total group? Please clarify.

Line 118: typo in 'Acliestorhinus'

Line 129: there are two abbreviations for 'exoccipital' ('eo' is used in figure 2b)

Line 164: 'Macdougall' should be 'MacDougall' (here and elsewhere in text)

Line 178: one can estimate skull length, or measure the exact length of a skull, but not 'estimate' the exact skull length

Line 306: change 'cuboidal' to 'quadrangular'

Line 311: each gnathostome has a single mandible (there is no such thing as a right mandible in tetrapod biology, unless you are describing a teratology); see also lines 435 and 909

Line 314: delete one of the full stops following 'nov'. See also lines 547, 589

Line 322: change 'however, no sutures can be discerned' to 'but no sutures can be discerned'

Line 330: change 'which is likely accurate' to 'with which we agree'

Line 421: excuse my pedantry, but the Latin 'draco' is from ancient Greek 'drakon', which means 'serpent' and was used to refer to snakes of unusual size ('ophis' was used for small snakes). That is to say, the ancient Romans and Greeks did not associate their terms with (mythological) winged reptiles, as this is a later concept (the English 'dragon' is derived from 'draco'). I am not suggesting a name change for the authors' new genus, but they should be clear on the etymology; authors may wish to check out the book "Dragon Myth and Serpent Cult in the Greek and Roman Worlds" by Daniel Ogden for background.

Line 436: typo in 'Riesz'

Line 449: change 'are not able to be identified at present' to 'are not identifiable at present'

Line 461: change 'recurvated' to 'recurved'

Line 657: a reference is needed here

Line 670: I have worked on extinct reptiles for many years, and I find the phrase 'more conventional reptilian bauplan' puzzling. Please elaborate on this concept.

Line 781: author name is 'deBraga'

Lines 790, 793, and 800: information missing from these references

Figure 2: label C missing from skull reconstruction

Reviewer: 2

Comments to the Author(s)

This is an interesting paper that extends the lineage of one important clade of Permian reptiles. The authors have nicely demonstrated that *Carbonodraco* is a parareptile.

I request the authors to use the term *Reptilia*, this name clearly has priority over the unnecessary "*Sauropsida*". See Modesto and Anderson (2004) for a phylogenetic definition of *Reptilia*.

I made some minor comments and corrected some typos (see attached file). The names *Acleistorhinus* and *acleistorhinids* are misspelled throughout the entire text. Please also check my comments to the illustrations in that file.

Reviewer: 3

Comments to the Author(s)

This paper re-describes some interesting specimens representing early eureptilians and parareptiles, which are important for understanding the origin and evolution of *Reptilia*. The descriptions are thorough, and my comments mainly refer to the terminology used to describe the teeth:

Line 99: "crafted by Donald Baird" sounds like he made it up. Maybe say "... we were able to study original latex peels taken/collected/made by DB and casts of..."

Line 156: "maxilla bears multiple canine-like peaks". What does this mean? Are all the teeth caniniform in shape? Using the word 'teeth' instead of peaks would make much more sense here, if this statement is actually referring to teeth and not some peaks made of bone on the maxilla. If this statement is referring to tooth size, maybe say: Maxillary teeth irregular in size along toothrow/ maxillary tooth height inconsistent along toothrow/ or something like that. See comments below regarding the use of 'canine' and separating tooth position, tooth size, and tooth shape in descriptions.

Line 188: "There a thin and delicately built anatomy is seen..." Add a comma after "There".

Line 195: The lacrimal

Line 199: The word "canines" in parentheses is not necessary here, since canine refers to a specific tooth type and tooth position in mammals. See comments below.

Line 215: Sides missing an S.

Line 320: confusing sentence “On the right, lingually preserved, dentary the symphysis...”

Line 444, 460, 462, 474: the use of ‘canine’ and ‘incisor’ is problematic, as these are mammalian terms referring to a specific location in the mouth and specific tooth shapes. Check out the discussion in Macdougall and Reisz 2014 on the use of the term ‘canine’ and ‘caniniform’ and switch your language accordingly (talk about size differences in term of tooth number, e.g., maxillary teeth 4 and 5 are twice the length of all other maxillary teeth). This will make scoring these animals in future phylogenetic analyses easier. Separate tooth size from tooth shape and tooth position.

MacDougall, M. J., and R. R. Reisz. 2014. The first record of a nyctiphruetid parareptile from the Early Permian of North America, with a discussion of parareptilian temporal fenestration. *Zoological Journal of the Linnean Society* 172:616-630.

Line 449: “are not able to be identified at present” can be “are not identifiable at present”

Line 461: “gently recurvated” should be gently recurved

Line 542: ‘...have places of 19 teeth...’ should be ‘for’ instead of ‘of’

Line 588: ‘... narrowing morphology that end with a...’ missing s

Line 650: Pardo et al. (2019) missing from reference list.

Lines 651-655 are very confusing and should be re-written for clarity, especially ‘Upland ecosystems are no more abundant in early amniotes than other ecosystems’

Line 657 (reference)

Line 673, 679 ‘multiple canine peaks in the maxillary tooth row’. See comments above. Separate tooth shape from tooth size and tooth position.

Line 678-680: Can you cite something that you would support your interpretation of this type of feeding?

Line 698: ‘had arboreal habits’ should be ‘was arboreal’

Figure 2: The peel doesn’t offer much information. Is it possible to make the drawing bigger in order to better see the described anatomy? This would make the description much easier to follow (especially the palate). The scleral ossicles should be included in the cranial reconstruction.

Figure 3: This figure doesn’t show anything informative. Maybe combine with figure 4, but maintain the detail and size of the interpretive line drawing? Consider putting photos of peels as supp info and just focus on the detailed drawings for the figures in the paper?

References: most species names missing italics

Author's Response to Decision Letter for (RSOS-191191.R0)

See Appendix B.

Decision letter (RSOS-191191.R1)

22-Oct-2019

Dear Mr Mann,

I am pleased to inform you that your manuscript entitled "Carbonodraco lundii gen et sp. nov., the oldest parareptile, from Linton, Ohio, and new insights into the early radiation of reptiles." is now accepted for publication in Royal Society Open Science.

Kind regards,
Lianne Parkhouse
Editorial Coordinator
Royal Society Open Science
openscience@royalsociety.org

on behalf of Dr Julia Brenda Desojo (Associate Editor) and Kevin Padian (Subject Editor)
openscience@royalsociety.org

Appendix A**ROYAL SOCIETY
OPEN SCIENCE****Reappraisal of Carboniferous early reptiles from Mazon
Creek and Linton reveals the oldest parareptile:
Carbondraco lundi gen. et sp. nov.**

Journal:	Royal Society Open Science
Manuscript ID	RSOS-191191
Article Type:	Research
Date Submitted by the Author:	08-Jul-2019
Complete List of Authors:	Mann, Arjan; Carleton University, Earth Sciences McDaniel, Emily J.; Carleton University, Earth Science McColville, Emily R.; Carleton University, Earth Sciences Maddin, Hillary; Carleton University, Earth Sciences
Subject:	palaeontology < BIOLOGY, taxonomy and systematics < BIOLOGY, evolution < BIOLOGY
Keywords:	Carboniferous, Mazon Creek, Eureptilia, Parareptilia, Amniota, Pennsylvanian
Subject Category:	Biology (whole organism)

**Author-supplied statements**

Relevant information will appear here if provided.

***Ethics***

*Does your article include research that required ethical approval or permits?:*

This article does not present research with ethical considerations

*Statement (if applicable):*

CUST_IF_YES_ETHICS :No data available.

***Data***

*It is a condition of publication that data, code and materials supporting your paper are made publicly*
*available. Does your paper present new data?:*

My paper has no data

*Statement (if applicable):*

CUST_IF_YES_DATA :No data available.

***Conflict of interest***

I/We declare we have no competing interests

*Statement (if applicable):*

CUST_STATE_CONFLICT :No data available.

***Authors' contributions***

This paper has multiple authors and our individual contributions were as below

*Statement (if applicable):*

40 A.M. designed the study, A.M. E.R.M E.J.M. collected and analysed the data, A.M. and H.C.M.
wrote the paper.

TITLE PAGE

Reappraisal of Carboniferous early reptiles from Mazon Creek and Linton reveals the oldest parareptile: *Carbondraco lundi* gen. et sp. nov.

Arjan Mann*, Emily J. McDaniel, Emily R. McColville, and Hillary C. Maddin

Department of Earth Sciences, Carleton University, 1125 Colonel By Drive, Ottawa, Ontario
K1S 5B6, CanadaRunning title: The oldest parareptile *Carbondraco lundi**Corresponding author: Arjan Mann; Department of Earth Sciences, Carleton University, 1125
Colonel By Drive, Ottawa, Ontario K1S 5B6, Canada; email: arjan.mann@carleton.ca

ABSTRACT

The Pennsylvanian-aged (309-307 Ma) Francis Creek Shale, Mazon Creek Lagerstätte, produces some of the earliest fossils of the major tetrapod lineages. The Mazon Creek record of the early ‘protorothyridid’ reptile, *Cephalerpeton ventriarmatum* (YPM 796), is known from a single part concretion containing a well-preserved anterior portion of a skeleton. Aside from the reptile *Hylonomus lyelli* from the slightly older deposits of Joggins, Nova Scotia, the remains of *Cephalerpeton* are the amongst the oldest known amniote fossils. *Cephalerpeton* cf. *C. ventriarmatum* has also been identified in the tetrapod fauna from Linton, Ohio, represented by a disarticulated cranium (CM 23055) and a single right dentary (BMNH R. 2667). Here we re-describe the anatomy of *Cephalerpeton* from all of its known fossils from Mazon Creek and the slightly younger coal deposits of Linton, Ohio, and additionally, describe new material from the latter. Our results indicate major anatomical differences between fossils of *Cephalerpeton* from Mazon Creek and those from Linton, likely representing new taxonomic diversity. The holotype is reconfirmed as a basal eureptile sharing close postcranial skeletal similarities to other protorothyridids, such as *Anthracodromeus* and *Paleothyris*. The skull of the holotype is long and lightly built, with large orbits, and a dorsoventrally short mandible similar to most basal eureptiles. This strongly contrasts the condition seen in *Cephalerpeton* cf. *C. ventriarmatum* from Linton where the cranial and mandibular elements appear proportionally taller. Additionally, the anteroposteriorly narrower and dorsoventrally taller maxilla of the Linton specimen is reinterpreted as excluding the lacrimal from the naris, similar to the condition seen in ‘short-faced’ parareptiles such as *Colobomycter* and *Acleistorhinus*. We propose a parareptilian affinity for the reptile remains from Linton, among the **acliestorhinids**, which makes it the oldest known example of a parareptile.

INTRODUCTION

Amniotes can be divided into two major lineages, synapsids (mammals and their extinct relatives), and sauropsids (crocodiles, birds, lepidosaurs and their extinct relatives). The origin and early diversification of these groups is believed to have occurred sometime during the Carboniferous, with the oldest known amniotes, *Hylonomus lyelli* and the putative synapsid remains of *Protoclepsydrops haplous* being found in the Bashkirian deposits of Joggins, Nova Scotia, Canada (313–316 Ma)(Allen *et al.*, 2014). The early records of reptiles come from the historic Pennsylvanian-aged coal measure localities. These include *Hylonomus* from Joggins, Nova Scotia, *Paleothyris* from Florence, Nova Scotia, *Brouffia* and *Coelostegus* from Nyrany, Czech Republic, *Anthracodromeus* from Linton, Ohio, and *Cephalerpeton* from Mazon Creek, Illinois, and Linton, Ohio. These fossils were originally comprehensively described as close relatives of reptiles by Carroll and Baird (1972) within the now defunct stem-reptile family Romeriidae, and subsequently considered to be part of a generalized Permo-Carboniferous assemblage of reptiles known as Protorothyrididae (Carroll, 1982; Heaton and Reisz, 1986; Reisz, 1997). The most recent attempt to place these fossils in a phylogenetic context was conducted by Müller and Reisz (2006). In this analysis Protorothyrididae was found to be paraphyletic, consisting of an array of basal eureptiles; notably, a clade containing *Protorothyris*, *Anthracodromeus*, and *Cephalerpeton* was found to form a sister taxon relationship with Diapsida.

One of these earliest known reptiles, *Cephalerpeton ventriarmatum* Moodie 1912, was described from a single part concretion from the Mazon Creek, Francis Creek shale, that included an articulated anterior half of a skeleton. Moodie (1912) briefly described this animal as a microsaurian tetrapod, which at this time was a collection of small-bodied reptilian tetrapods. Due to the brevity of Moodie's description, the anatomy of *Cephalerpeton* was redescribed by Gregory (1948), and again by Carroll and Baird (1972). The latter description was the first to group *Cephalerpeton* with other early reptiles. Further anatomical work by Reisz and Baird (1983) introduced additional remains of *Cephalerpeton* from Linton, Ohio. These remains consist of a macerated skull, CM 23055, and a referred dentary that bore somewhat aberrant anatomy from that of *Cephalerpeton ventriarmatum* from Mazon Creek, YPM 796. Reisz and Baird (1983) referred the Linton material to *Cephalerpeton* cf. aff. *C. ventriarmatum*.

Unfortunately, since these descriptive works, *Cephalerpeton* has rarely been included in
phylogenetic analysis of reptiles or amniotes, despite being important to the origin and evolution
of Reptilia. Updated comparative osteological analyses, in light of recent advances, are required
in order to include as many of these taxa as possible, and facilitate revised systematic studies of
the origin of reptiles. Here, we provide new, comprehensive, and comparative descriptions of all
known specimens of *Cephalerpeton* from Mazon Creek and Linton in order to provide an
updated osteological assessment of the taxon for future phylogenetic analyses. In our study, we
were able to study original latex peels (crafted by Donald Baird) and casts of *Cephalerpeton*
*ventriarmatum* from Mazon Creek in order to supplement anatomical analysis on the damaged
holotype specimen. We were able to recognize new craniodental features of *Cephalerpeton*
*ventriarmatum* which distinguish it from the Linton material, which we assign to the new
**acliestorhinid** parareptile, *Carbonodraco lundi* gen et sp. nov. As a result, *Carbonodraco lundi*
represents the oldest known parareptile.

MATERIALS AND METHODS

Specimens were studied at: American Museum of Natural History (AMNH), New York;
Carnegie Museum of Natural History (CM), Pittsburgh; Field Museum of Natural History
(FMNH), Chicago; Smithsonian Institution (USNM), Washington DC; and Yale Peabody
Museum (YPM), New Haven. Additional comparative specimens from the British Museum of
Natural History (BMNH), London, Museum of Comparative Zoology (MCZ), Cambridge, and
Museum für Naturkunde (MB), Berlin, were compared based on casts, latex peels, and existing
literature. As a result, the combined comparative dataset constituted nearly all known basal
eureptiles from the Carboniferous, including notable material from the Francis Creek Shale of
Mazon Creek, Illinois, and Linton, Ohio, as well as material from the cannel coal below the
Lower Kittanning coal of Five Points, Ohio. We were also able to study the following
parareptiles: *Delorhynchus*, *Colobomycter*, ***Acliestorhinus***, and *Erpetonyx*. Photography was
conducted using a Sony Alpha ILCE 5000 camera with a F3.5 macro lens.

Illustration of YPM 796 was redrawn and modified from Carroll and Baird (1972).
Illustration of CM 23055 was redrawn and modified from Reisz and Baird (1983). Anatomical
illustrations of CM 81536 and CM 41714 were drawn from original specimens and casts or

123 peels. All figure drawings were generated and formatted in Photoshop CS6 (Adobe, San Jose,
CA).

**Anatomical abbreviations**

**aa**=atlantal arch; **ac**=anterior coracoid; **an**=angular; **ar**=articular; **axa**=axis arch;
**axp**=axis pleurocentrum; **bo**=basioccipital; **c**=clavicle; **cth**=cleithrum; **dv**=dorsal vertebrae;
**d**=dentary(**dr/dl**=right/left); **ect**=ectopterygoid; **eo**=exoccipital; **ex**=exoccipital; **f**=frontal;
**gs**=gastralia; **h**=humerus; **l**=lacrimal; **j**=jugal; **m**=maxilla; **mc**=metacarpals; **n**=nasal; **p**=parietal;
**pf**=postfrontal; **pmx**=premaxilla; **pp**=postparietal; **prf**=prefrontal; **pl**=palatine; **pt**=pterygoid;
**pro**=proatlas; **q**=quadrate; **qj**=quadratojugal; **r**=radius; **s**=scapula; **scl**=scleral ossicles;
**sm**=septomaxilla; **so**=supraoccipital; **sq**=squamosal; **sr**=surangular; **st**=stapes; **sv**=sacral
vertebra; **sp**=splenial; **t**=tabular; **tfpt**=transverse flange of the pterygoid; **u**=ulna; **v**=vomer.

**Systematic Palaeontology**

Tetrapoda Jaekel 1909

Amniota Haeckel 1866

Eureptilia Olson 1947

*Cephalerpeton* Moodie 1912

*Cephalerpeton ventriarmatum* Moodie, 1912

(Figs. 1 and 2)

**Hototype:** YPM 796, anterior portion of a skeleton, including the upper limbs, cranium and
lower jaws.

**Locality and Horizon:** Mazon Creek, Grundy County, Illinois, USA. Francis Creek Shale,
above the Morris (no. 2) Coal, Carbondale Formation, Middle Pennsylvanian (Moscovian).

**Revised Diagnosis:** A basal eureptile diagnosed by the following autapomorphies: 16 wide-
based conical teeth in maxilla; maxillary dentition significantly enlarged compared to dentary
teeth; maxilla bears multiple canine-like peaks; palatal bones covered in a shagreen of denticles.
Shares a slender, rod-like ulna and radius with *Anthracodromeus*, *Paleothyris* and basal diapsids.

**Comments:** Upon the removal of *Carbonodraco lundi* gen. et sp. nov. (formerly *Cephalerpeton*
cf. aff. *C. ventriarmatum*) from Linton, Ohio, *Cephalerpeton ventriarmatum* is only known from
YPM 796, and currently restricted to the Mazon Creek **lagerstätte**. One similarity between
*Cephalerpeton* and *Carbonodraco* is the presence of plicidentine and intense enamel fluting on
the tooth crowns. This trait is also widely shared with other parareptiles and synapsids
(Macdougall et al., 2015). Because of this wide dispersal of the trait it is not included in the
diagnosis, although it may be unique to *Cephalerpeton* among ‘protorothyridid’ reptiles at least.

DESCRIPTION

**Cranial anatomy of *Cephalerpeton***

The skull of YPM 796 is crushed and incompletely preserved (Figs 1–2). Many of the
cranial elements are preserved in ventral view, revealing their internal anatomy. Pieces of the
palate are disarticulated throughout the skull. Both mandibles are shifted to the anatomical right
side of the cranium. In general, the right side of the cranium is better preserved, showing a nearly
complete cheek region that is absent from the left side of the skull. The cranial anatomy of
*Cephalerpeton* is overall lightly built, with the elements being thin and with the skull build being
quite narrow. This compares well with contemporaneous Carboniferous ‘protorothyridids’, early
captorhinids such as *Euconcordia*, as well as later occurring diapsids such as *Araeoscelis*,
*Petrolacosaurus*, and *Spinoequalis*. Because the premaxillae are not well preserved, estimation
of the exact skull length is not possible.

Antermost on the skull are preserved remnants of a single left premaxilla. The
premaxilla is lightly built and has spaces for at least three teeth. Unlike the reconstruction of

Gregory (1948), we interpret the premaxilla as bearing a slightly recurved dorsal ascending
process, similar to that of other early eureptiles such as ‘protorothyridids’ and araeoscelids,
however, not as hooked as that in captorhinids or recumbirostrans (Carroll and Baird, 1972). The
dorsal ascending process of the premaxilla has an elongate morphology. The lateral contact
between the premaxilla and the maxilla is not preserved in any of the peels or casts examined.

Two maxillae are preserved in YPM 796, both revealing their medial surfaces. The left
maxilla is the most completely preserved. There a thin and delicately built anatomy is seen,
similar to that of other basal eureptiles such as *Hylonomus*, *Palaeothyris*, *Thuringothyris* and
most captorhinids. Anteriorly the bone descends to a thin short anterior process that narrowly
contributes to the posterior edge of external naris. Just anterior to the orbit, the maxilla ascends
to a low facial lamina similar to that of most protorothyridid reptiles. There is a thin long
posterior process that terminates approximately below the posterior margin of the orbit. It
appears this area of the maxilla was excluded from the orbit by a thin connection between
lacrimal and jugal. The left maxilla preserves 16 tooth positions with 14 teeth in place in YPM
796. Each tooth consists of a wide cone, which bears crenulations on the crown likely
representing enamel fluting and some of the teeth show resorption pits **a** their bases. The teeth of
YPM 796 express a degree of heterodonty separating it from most other early ‘protorothyridids’.
The largest teeth in the tooth row are located directly under the facial lamina (canines), however,
an even larger tooth is located under the anterior margin of the orbit. This means there is no
single, distinct ‘canine’ region, but rather at least two peaks in maximum height along the tooth
row are present. There is also no noticeable taper in tooth height posteriorly unlike the condition
seen in CM 23055 (*Cephalerpeton* cf. *C. ventriarmatum*).

The lacrimals are present on both sides of the cranium and preserved in medial aspect.
The lacrimal morphology is shared with most ‘protorothyridid’ reptiles, as well as araeoscelids,
such as *Petrolacosaurus*. The right lacrimal is obscured by the vomer, however, the left is nearly
perfectly represented. From the left side the lacrimal is seen to be a long element, meeting the
naris anteriorly, and contributing to the anterior orbital margin at its posteriormost extent. The
left lacrimal also reveals a long lacrimal canal represented by a raised tube terminating at an
opening just posterior to the narial opening. The length of the lacrimal canal is significantly
longer than that of CM 23055 (*Cephalerpeton* cf. *C. ventriarmatum*) from the Linton locality.
Posteriorly, on the left lacrimal there is a recess for the ventrally directed process of the

adjoining prefrontal to nest. Both lacrimals preserve a long posteroventrally extending process
that meets the anterior process of the jugal, as described by Carroll and Baird (1972).

The prefrontals are present on both side of the cranium. The prefrontal is excluded from
the external naris by the contact between the nasal and lacrimal on the lateral surface of the
snout. The prefrontal is roughly Y-shaped with prominent anterior, ventral, and posterior
processes. These processes are part of a series of processes cascading off of nearly every
circumorbital element. This pattern is shared widely among early reptiles, including parareptiles.
The anterior process of the prefrontal is extensive, covering a large portion of the lateral snout.
The posterior process is quite long and thin, similar to that of other ‘protorothyridids’. The most
striking feature of the prefrontal is the extremely long, ventrally directed process, which forms
the majority of the anterior margin of the orbit. This process is delicately built and fits into a
facet on the lacrimal.

The postfrontal is a falciform-shaped element, present on either side of the cranium. In
YPM 796 the postfrontals are also preserved in ventral aspect. The postfrontal has a somewhat
long anterior process, and a rather short extension on to the cheek. This is unlike the condition in
more derived diapsids, but similar to the condition seen in other basal eureptiles like *Hylonomus*
(Carroll and Baird, 1972).

The nasals are preserved in medial aspect, where they are seen to have a long, thin
subrectangular morphology. The right nasal is slightly overlapping the left (Fig. 1). There is a
slight anterior expansion directed toward the narial margin away from the midline. The
anteriormost end of the nasals taper to a point, creating a midline internasal recess that likely
accommodated the thin elongated ascending dorsal process of the premaxilla.

The frontals are preserved in a similar manner to the nasals in being slightly overlapping,
with the right better exposed. They are long rectangular elements that show a small contribution
to the dorsal orbital margin. This is indicated by a pinching laterally in mid-region of the frontal.
Overall, both the frontals and nasals are extremely narrow, as noted by previous studies
(Gregory, 1948; Carroll and Baird, 1972). The anterior margin of the frontal appears to form a
straight sutural contact with the nasal. Because the parietals have shifted out of place and are not
well-preserved it is not possible to determine the nature of the sutural contact between the
frontals and parietals.

The parietals are represented in YPM 796 by two large semi-lunate ossifications. Both
appear incomplete and possibly would have been better represented on the unknown counterpart.
Based on the somewhat short space between the occipital elements and the frontals it is assumed
the parietals were anteroposteriorly short and only moderately expanded lateromedially, keeping
with the narrow shape of the cranium as a whole.

The cheek region on YPM 796 is comprised of a jugal, postorbital, squamosal and
quadratojugal. These are all represented only on the right side (except for the squamosal),
preserved in medial view. Anteriorly, the jugal forms a thin process that contributes to the
posteroventral margin of the orbit. This process meets, and possibly slightly overlapped, the
lacrimal. The cheek region of the jugal is moderately expanded. There is a small notched contact
dorsally where the jugal would have accepted the postorbital and a straight contact posteriorly
with the squamosal. The postorbital is not well represented on the right side, where it is
overlapped by the squamosal. From what is visible, taking into account the relationship of the
other cheek elements, the postorbital is likely a small quadrangular element extending from the
top of the orbital margin, adjacent to the postfrontal, ventrally towards the jugal, and contacting
the squamosal posteriorly. It appears the small width of the postorbital is responsible for the
short appearance of the postorbital region. The squamosal forms the majority of the lateral cheek
and is represented by a large plate-like element on the right side of the cranium, as well as a
displaced left squamosal that is present near the left forelimb. The squamosal is taller than it is
wide, and indicates a relatively high cheek region. The contacts with most other cheek and dorsal
skull elements are not clear. On the right side a relatively straight, ventral contact with the
quadratojugal can be observed. The quadratojugal itself is not well preserved due to the
posteriormost region falling off of the original fossil. The right quadratojugal is preserved in
medial aspect like the rest of the cheek. It is roughly triangular in shape with the wider end being
posteriorly located. Posteriorly, the quadratojugal is partially overlapped by the quadrate. The
right quadrate can be seen to be a long, cylindrical element. It is incomplete anteriorly, missing
the process that would meet the pterygoid. Ventrally the articular surface for the mandible is well
preserved, showing a median depression between two condyles.

Because the skull is preserved in its ventral aspect, the palate is well represented;
however, its elements are disarticulated throughout the skull. The preserved elements of the
palate include both vomers, the right ectopterygoid, the right pterygoid, and the right palatine.

All of these elements are covered in a shagreen of small denticles. The vomer is represented
anteriorly in the skull and is a quadrangular element with few distinctive qualities. Both left and
right vomers appear to be tapered anteriorly, creating a space that likely accommodated the
internal median processes of the premaxillae. The contacts with other elements can only be
speculated upon, but it is likely that the vomer met the palatine posterolaterally, the pterygoid
medially, and the ectopterygoid posterolaterally. A single partial right palatine is present as a
wide quadrangular element, which bears an emargination on the anterolateral surface where the
internal naris would reside. The right ectopterygoid is represented by an extremely thin,
denticulate element. It is likely the element was slightly wider than this in life based on the
apparently broken margins of the bone. The right pterygoid is well represented, only missing its
quadrate ramus. A prominent anterior ramus of the pterygoid is seen in ventromedial aspect,
bearing a shagreen of denticles. It is possible that the anterior ramus of the pterygoid is slightly
raised to a small boss, similar to that of other basal eureptiles, but this is difficult to determine
given the lack of association with the other palatal elements. The most striking feature of the
pterygoid is the well-developed transverse flange, which occurs at an approximately 90 degree
angle to the midline margin of the anterior ramus. The teeth on the transverse flange are not well
preserved on any peel, but a field of slightly larger teeth can be observed on the resin cast and
latex peels.

The occiput is not well represented in YPM 796; however, a few elements are preserved
including the supraoccipital, a partial basioccipital, and at least one exoccipital. The
supraoccipital is a large plate-like element that is roughly butterfly-shaped. The supraoccipital
forms the dorsal margin of the foramen magnum. The middle of the supraoccipital is slightly
raised forming a small, sagittally-oriented ridge. Underlying the supraoccipital is an amorphous
element identified as likely being the basioccipital, as identified in Carroll and Baird (1972).
Little can be said about this element other than it is slightly concave in dorsal aspect. At least one
exoccipital, probably the left, is present and disarticulated, being located near the atlas and axis.
It is an elongated, cylindrical element. A medially located depression on the exoccipitals likely
represents the location of the jugular foramen. An element identified by Carroll and Baird (1972)
as the atlas pleurocentrum may in fact be the other exoccipital; however, it is not preserved well
enough to be confidently identified as such.

Lastly, there is a series of 7–9 scleral ossicles. Scleral ossicles are readily preserved in
Mazon Creek tetrapods. Those observed in temnospondyls from Mazon Creek are often small,
cuboidal elements numbering around 22–24. Those observed in YPM 796 differ in being large,
rectangular ossicles that more closely resemble those found in other amniotes such as the
captorhinid *Reiszorhinus*.

**Mandible of *Cephalerpeton***

Preservation of the mandibles of YPM 796 includes the entire right mandible in medial
perspective and the left dentary in lateral aspect (Fig. 2). The mandibles, in general, display a
very gracile morphology. This is in contrast to the elements found in the specimens assigned here
to *Carbonodraco lundi* gen et sp. nov.. The lateral surface of the dentary shows a pitted and
slightly rugose ornamentation. This is interesting because most of the cranial elements on
*Cephalerpeton* are preserved revealing their interior surfaces, thus cranial ornamentation remains
largely unknown. Rugose cranial ornamentation is a significant feature of the Captorhinidae,
present in most of their members with the exception of *Opisthodontosaurus*.

The left and right dentaries are thinly built with an unexpanded symphysis that tapers
gradually anteriorly. On the right, lingually preserved, dentary the symphysis appears even
thinner, and is excavated towards the centre. There may be a splenial attached to the right
dentary, however, no sutures can be discerned. Neither dentary preserves their coronoid
processes. The right surangular, angular, and articular are preserved revealing their medial
surfaces. These postdentary bones make up over one third of the length of the lower mandible.
The surangular is the best represented of the postdentary bones. It is an irregular, oval-shaped
element in lateral aspect. The angular is represented by an elongated element in medial view, and
the articular is a small, oval-shaped ossification at the posteriormost end of the mandible.

The dentition on the dentaries differs slightly from that of the upper jaws. The left
dentary has 18 teeth in place with spaces for a few more teeth. Carroll and Baird (1972)
estimated 24 teeth, which is likely accurate. The dentition on the lower jaws consists of sharply
pointed conical teeth that are recurved apically. At the anteriormost end of the dentary the
dentition is slightly enlarged and anteriorly directed similar to that of most ‘protorothyridid’
reptiles. Overall the dentition on the dentaries appears less wide and tall than that on either
maxilla.

Postcranial anatomy of *Cephalerpeton*

It is worth noting that our interpretation of the postcranial anatomy is largely consistent
with the detailed descriptions provided by Carroll and Baird (1972). Here we briefly overview
the anatomy and provide a few updated comparisons. The postcranial skeleton consists mostly of
the presacral vertebral column (including the atlas-axis vertebrae), dorsal ribs, ventral gastralia,
pectoral girdle elements, and the forelimbs. The vertebral count proposed by Gregory (1948) of
25–26 vertebrae was identified by Carroll and Baird (1972) as erroneously including elements of
the occiput into the cervical vertebral series. Whereas 23 presacral vertebrae can be identified,
we agree with Carroll and Baird (1972) that a 28 total presacral vertebrae count is plausible.

The proatlas is represented in *Cephalerpeton* by a small round ossification in between the
occipital elements and the atlas vertebra. The atlantal vertebral components cannot be
confidently identified, however, a number of elements were identified by Carroll and Baird
(1972) including the atlantal arch as well as the atlas pleurocentrum (here possibly considered an
exoccipital instead). We also note that unidentified elements (unlabelled in Fig. 2) adjacent to the
right exoccipital may also represent elements of the atlas, such as the atlas intercentrum. The
small cylindrical axial pleurocentrum and large somewhat fan-like axial arch are present. It
appears as though the arches are fused to the pleurocentra throughout the vertebral column due to
the tight association of the two. The observed line running between the two may simply be a
crack. The neural arch morphology throughout the vertebral column remains consistent, with
strongly overlapping and well developed prezygapophysis and postzygapophysis. The neural
arches also bear a small anterior excavation that is depressed into the base of the somewhat tall
neural spine. The neural spines are only fully preserved on approximately the eight anteriormost
vertebrae. There they can be seen to have somewhat rounded margins, possibly indicating they
were weakly ossified. Their height is comparable to early reptiles such as *Anthracodromeus*, but
also some varanopid synapsids, and other early pelycosaurian-grade tetrapods. A small
transverse process can also be observed on some of the vertebrae. The pleurocentra are formed
by large elongated cylinders that are concave on the lateral surfaces, and do not bear any
noticeable keels or ridges. Ventrally, wedged between each pleurocentrum is a small, likely
crescent-shaped, intercentrum. The dorsal ribs are well-represented along the postcranial
skeleton. All ribs are single headed. The cervical ribs have a short but thick bar-like morphology,

which develop slight recurvature moving posteriorly down the series. The dorsal ribs posterior to
the pectoral girdle are the longest approximately three times the vertebral length. These ribs are
also thin and highly recurved. Towards the posterior end of the preserved column the ribs shorten
to a thin bar-like morphology, indicating the approach of the pelvic region.

The pectoral girdle is represented by the cleithrum, clavicle, scapula, and anterior
coracoid (procoracoid). Overall the pectoral girdle is lightly built like that of early basal
eureptiles such as *Anthracosaurus*, *Paleothyris*, *Hylonomus*, *Brouffia*, and early diapsids. The
left cleithrum, preserved anterior to the other pectoral elements, is teardrop shaped, and slightly
concave indicating it is the interior surface. The clavicle is roughly horn-shaped, widening
medially towards where the head of the interclavicle would have been. The left clavicle appears
to have a somewhat short lateral process based on the size of YPM 796, however, this clavicle
may be incomplete. A piece of the right clavicle was identified by Carroll and Baird (1972) as
residing adjacent to the left humerus, however, it is possible that this element is something else
and here is considered to be unidentifiable. A partial right scapula is also preserved revealing its
exterior surface, as well as the anterior coracoid. The anterior coracoid is ovoid in outline and
slightly overlapped by the scapula on the posterolateral margin.

The right and left forelimbs are well preserved in YPM 796, only missing the phalanges
and most of the carpal elements. Overall the long morphology of the forelimb elements closely
resembles the forelimbs of early diapsids, such as *Spinoequalis*, *Petrolacosaurus*, and
*Araeoscelis*, but differs from the squat and robust limb morphology of *Thuringothyris*,
captorhinids, and recumbirostrans. Both humeri are preserved, their anatomy shows a long rod-
like shaft with a small proximal end with a moderately developed head. The distal end is greatly
expanded. Both proximal and distal ends appear to be rotated at about 90 degrees to one another.
The left distal humerus is rotated to reveal its articular surface, showing a shallow capitulum, and
weakly developed entepicondyle. As pointed out by Carroll and Baird (1972), no record of any
ridges or supinator process can be found on the humerus. This may indicate immaturity in the
animal.

The ulna and radius are represented in both forelimbs as long rod-like elements that are
approximately equivalent in length and only slightly shorter than the humerus. This is shared
with *Spinoequalis*, *Petrolacosaurus*, and *Araeoscelis*. The proximal and distal ends of these
elements appear both small in width and poorly developed. The ulna of YPM 796 has no

developed olecranon process, unlike that of captorhinids such as *Ophisthodontosaurus* (Reisz et
 al., 2015). Five metacarpals are preserved on the right manus and two on the left manus. They
 are also long and rod-like, the greatest of which is half the length of the zeugopodial elements,
 likely indicating the manus was long, similar to that of *Anthracodromeus*. The right manus also
 shows 2–3 overlapping distal carpals preserved that are roughly cuboid in shape.

One of the unique features of YPM 796 is the presence of a pelage of ventral gastralialia, as
 well as soft-tissue, integumentary impressions hugging the forelimbs. The gastralialia are thin
 elongate rods that are canted anteriorly to meet at the midline, together forming a chevron shape.
 The gastralialia present in *Cephalerpeton* are of the ‘reptilian’ morphology in that they are thin and
 unornamented (no concentric growth lines), and thus are similar to those found in *Hylonomus*,
 *Anthracodromeus*, but also in varanopids, and some recumbirostrans.

SYSTEMATIC PALEONTOLOGY

Tetrapoda Jaekel, 1909

Amniota Haeckel, 1866

Parareptilia Olson, 1947

Acleistorhinidae Daly, 1969

*Carbonodraco* gen. nov.

(Figs. 3–5)

**Etymology:** Generic name derived from the latin words ‘*Carbo*’ (coal), and ‘*Draco*’ for dragon.
 Specific name is in honor of Dr. Richard Lund whom found the holotype.

**Diagnosis:** As for the type and only species.

*Carbonodraco lundi* sp. nov.

=*Cephalerpeton* cf. *C. ventriarmatum* Reisz and Baird 1983

4284

**Holotype:** CM 23055 consisting of a slightly disarticulated skull including both maxillae and
associated dentition, left premaxilla, right lacrimal, left prefrontal, left parietal, left frontal, both
dentaries and associated dentition, splenials, left surangular, and vomers.

**Referred Material:** CM 81536 consisting of a pair of dentaries preserved in lingual perspective
(collected by Scott Mckenzie), BM(NH) R. 2667 (J. W. Davies Linton Collection) a right
mandible in lingual perspective (Figure 2: Riesz and Baird, 1983).

**Locality and Horizon:** Linton Diamond Mine, Saline Township, Jefferson County, Ohio, USA.
Discovered and collected by Richard Lund and party in 1972 from within the cannel coal below
coal seam identified as the Upper Freeport Coal, Allegheny Group, Middle Pennsylvanian.

**Diagnosis:** An **acliestorhinid** parareptile diagnosed by the following unique combination of
characters: vomers covered in a shagreen of denticles; parietals wide; pineal foramen anteriorly
located. Two enlarged canine teeth on maxillae and anteriormost premaxillary tooth enlarged to
the height of maxillary canines shared with *Colobomycter*. Lacrimal short and excluded from
external naris shared with *Colobomycter* and *Acleistorhinus*. High facial lamina of the maxilla
and pitted ornamentation shared with **acliestorhinids**.

**Comments:** Some elements in the holotype CM 23055 are not able to be identified at present,
while others can only be tentatively identified (outlined in the description below). Hopefully with
new discoveries and ongoing research at the Linton locality, the cranial anatomy of

DESCRIPTION

A single left premaxilla is preserved in the holotype specimen of *Carbonodraco lundi*
gen et sp. nov., CM 23055. It appears to have a short lateral process and a high dorsal ascending
process. The lateral surface is ornamented with tiny foramina. The premaxilla bears spaces for

approximately three teeth, two of which are in place, including a significantly enlarged anterior
tooth that is equivalent in length to the canines on the maxillae. The teeth on the premaxilla,
including the enlarged tooth, are gently recurvated towards the apex. Along the base of the
enlarged incisor there can be seen large grooves likely indicating the presence of plicidentine.
Overall the structure and size of the tooth is most similar to that of the parareptile *Colobomycter*
*vaughni*, whereas the tooth in *Colobomycter pholeter* is even larger (Macdougall et al., 2016).

CM 23055 preserves the right and left maxilla in lateral view (Figs. 3–4). The
ornamentation present on the lateral surface shows distinct, large pitting similar to that of other
known parareptilian taxa, including **acleistorhinids** such as *Colobomycter* and *Delorhynchus*. The
large pitting on the facial lamina of the maxillae is highly comparable to that of *Colobomycter*
*pholeter* and *Colobomycter vaughni* (Macdougall et al., 2016; 2017). The facial lamina of the
maxilla is tall and narrows slightly dorsally. The subnarial process of the maxilla bears an
anterolateral foramen adjacent to the naris similar to the maxilla of *Acleistorhinus*,
*Colobomycter*, and an assortment of other parareptiles. The posterior process tapers significantly
moving away from the facial lamina. The conical and sharply pointed tooth crowns display
substantial heterodonty in terms of size along the tooth row. Two canine teeth are significantly
larger than the rest, and are similar in size. Many of the teeth clearly show linear grooves
beginning at the tooth base and ending midway towards the apex of the tooth crown. The peel
has been slightly worn and this may be the reason why this feature is not present on all of the
teeth.

The lacrimal is represented only on the left side and is preserved in medial view in CM
23055. The lacrimal forms a portion of the anterior orbital margin. The lacrimal is comparatively
shorter than that in basal reptiles such as ‘protorothyridids’ and captorhinids, but also
comparatively shorter than that in many recumbirostran taxa (e.g. *Euryodus*). Although Reisz
and Baird (1983) reconstructed the lacrimal as being incomplete anteriorly, we find this unlikely
due to the presence of an anterior margin on both the latex peel and original specimen in
conjunction with the presence of a short lacrimal canal, which presumably like other reptiles
exits just prior to the anterior margin of the element (see Fig. 3; Macdougall et al., 2016). The
short morphology of the lacrimal resembles that of some acleistorhinid parareptiles (e.g.
*Acleistorhinus*, *Colobomycter*, and some specimens of *Delorhynchus*), where the lacrimal is
excluded from the naris, and partially overlapped by the high facial lamina of the maxilla

(Macdougall *et al.*, 2016). On the posterodorsal edge of the medial surface of the lacrimal is a
long groove that accepts the ventral process of the prefrontal. The posteroventral process of the
lacrimal is short, and dissimilar to that present in *Cephalerpeton* and other basal eureptiles, but is
close in morphology to that of *Colobomycter* and *Acliestorhinus*.

A single left prefrontal is preserved in CM 23055 also in medial aspect. This element is
roughly Y-shaped with all of the processes approximately the same length and width. The ventral
process of the prefrontal can be elegantly matched with the posterodorsal recess on the lacrimal.
This ventral process of the prefrontal, in conjunction with the posterior process of the prefrontal
form the anterodorsal margin of the orbit.

A nasal is tentatively identified in CM 23055, disarticulated and now located between the
vomers. The posterior margin of the nasal is overlapped by the right vomer, however, its anterior
margin is visible, revealing what appears to be the dorsal surface of the right nasal. There is an
anterolateral flange extending to what may be the anterior margin of the external naris.
Anteromedially there is recess which might have housed the ascending dorsal process of the
premaxilla.

The left frontal is preserved in ventral aspect and the right frontal may also be present,
overlapped by the anterior end of the left. The frontal morphology is both long and rectangular
with a small median lappet that likely formed a contribution to the dorsal margin of the orbit.
Overall the frontal is also quite wide, and both frontals together would have formed a wide
interorbital region. This is another feature that is drastically different from the condition seen in
*Cephalerpeton*, but is similar to that seen in acleistorhinid parareptiles.

The parietals are represented only by the left element that is preserved in ventral aspect.
The parietal is basically square shaped in outline. This element is very wide in comparison to
that of most protorothyridids, early eureptiles, and even most parareptiles. The anterior margin of
the parietal is straight indicating the suture with the frontal was likely simple and not strongly
interdigitated. The posterior margin of the parietal is emarginated slightly medially creating,
together with its antimere, a recess for median postparietals. The parietal-parietal contact is
straight with the exception of a small emargination for the anteriorly-located pineal foramen.
This anterior location of the pineal foramen is unique to *Carbonodraco* in comparison to other
*acliestorhinids*. The ventral surface is ornamented with striae, and bears a slightly raised medial
surface extending from the posterolateral edge.

The only palatal elements identifiable in CM 23055 are the vomers. The right vomer is
preserved in ventrolateral aspect, while the left vomer is only visible in lateral aspect. They both
show a high median lamina. The right vomer is better exposed, showing that the vomers are
roughly triangular in ventral aspect. The right vomer narrows anteriorly towards the internal
median premaxillary contact, and the vomer widens posteriorly towards both the palatine and
pterygoid. The vomer is covered in a shagreen of denticles. This is somewhat unique for a
parareptile where the vomers often bear a continuation of teeth from a denticulate boss on the
pterygoid (e.g. *Acleistorhinus*). However, the parareptile *Colobomycter pholeter* also has a
vomer with an uneven distribution of denticles, although not to the same degree as that seen in
CM 23055 (Macdougall et al., 2017; pers. obs. A. Mann).

The mandible of *Carbonodraco lundi* gen. et sp. nov. is represented by both dentaries,
two tentatively identified splenials and the left surangular preserved among the three specimens
(Figs. 4 and 5). The dentary is by far the best represented anatomical element in the material,
being known from the holotype specimen CM 23055, as well as the two referred specimens –
BM(NH) R. 2667 (reported by Reisz and Baird [1983]) and CM 81536 (a new specimen). The
dentary is quite robust, including at the slightly upturned symphysis. The lateral surface has
some gentle rugosity and small foramina anteriorly. Medially, there are fine ridges preserved that
may indicate the location of at least one coronoid ventral to a relatively low coronoid process.
The tooth counts are slightly variable between specimens of *Carbonodraco lundi* gen et sp. nov.,
however, they are within the range of variation seen in extant reptiles (Brown *et al.*, 2015). The
dentaries of CM 23055 bear an estimated 16–17 tooth positions, those of CM 81536 have places
of 19 teeth and the dentary of BM(NH) has 16 tooth positions. The teeth of CM 81536 bear a
weak degree of heterodonty, with the anteriormost dentary teeth being slightly larger than the
rest. Unlike *Cephalerpeton*, the dentary teeth of *Carbonodraco lundi* gen et sp. nov. are
approximately the same size, if not larger, than the opposing teeth on the maxillae and are of the
same morphology (not recurved). CM 81536 preserves details of the dentition better than any
other specimen of *Carbonodraco lundi* gen et sp. nov.. The morphology of each individual tooth
is conical and tapered more abruptly in the apical portion of the tooth crown. Each tooth crown
bears distinct enamel fluting consisting of very fine parallel grooves. The tooth bases and mid-
sections also show large grooves, often three or more. These larger grooves are interpreted as

plicidentine, similar to that found in parareptiles such as *Colobomycter*, as well as a variety of
other paleozoic tetrapods.

The splenial is a flat, long, tabular element with a posterior lappet that may have cupped
the posterior end of the dentary. The left splenial located adjacent to the left dentary appears to
be preserved showing its exterior surface, whereas the right splenial appears to be showing its
interior surface. The left surangular is partially preserved (it is overlapped by the right splenial).
It bears a moderately developed crest confluent with the coronoid process of the dentary, and a
slightly concave lateral surface.

SYSTEMATIC PALEONTOLOGY

Tetrapoda Jaekel, 1909

Amniota Haeckel, 1866

Reptilia indet.

**Material:** CM 41714 consisting of a left mandible including a dentary (broken into two pieces),
splenial, angular and surangular preserved in lateral view, rib fragments, and a partial centrum
(Fig. 6).

**Locality and Horizon:** Linton Diamond Mine, Saline Township, Jefferson County, Ohio, USA.
Discovered and collected by Richard Lund and party in 1972 from within the cannel coal below
coal seam identified as the Upper Freeport Coal, Allegheny Group, Middle Pennsylvanian.

**Comment:** Overall the mandibular morphology indicates a relatively long jawed reptile, unlike
*Carbonodraco lundi* gen et sp. nov., which is also found at the locality. Since there is no skeletal
overlap with *Anthracodromeus* it cannot be placed within that taxon either. Thus, we recognise
CM 41714 as a distinct, but indeterminate, reptile taxon in the Linton fauna.

**Description of CM 41714 (indeterminate reptile)**

CM 41714 includes a left mandible and associated postcranial fragments that bear some
resemblance to early reptiles (Fig. 6). The preserved mandible of CM 41714 includes a dentary
(the anteriormost portion of which is broken and displaced), splenial, angular, surangular and
articular preserved in lateral view. Postcranial fragments include a few rib fragments and a
partial centrum. Additionally, there is a large unidentifiable bone located on the edge of the
block beneath the anterior dentary fragment.

The anterior portion of the dentary shows a rapidly narrowing morphology that end with
a small unexpanded symphysis, unlike the dentary of *Carbonodraco lundi* gen et sp. nov.. The
posterior portion of the dentary is deep dorsoventrally and has a strongly convex ventral margin.
The lateral surface shows a slightly rugose ornamentation, with small foramina dispersed
throughout, similar to the ornamentation seen in some captorhinid eureptiles. In total, seventeen
teeth are preserved, with spaces for more at the posterior end. The teeth are homodont, with a
wide, bulbous base. The tooth crowns are highly worn, yet they retain a somewhat pointed apex.
Some captorhinids and recumbirostrans show a similar tooth morphology, however, the teeth are
not well enough preserved to adequately draw detailed comparisons. Unlike *Carbonodraco* or
*Cephalerpeton*, no evidence of enamel fluting or plicidentine grooves can be found on the teeth
of CM 41714.

The splenial is elongate, buttressing the ventral surface of the dentary. Its ventral margin
continues the convex shape of the entire mandible. The splenial also bears rugose ornamentation
and pitting similar to the dentary. Posteriorly, the splenial contacts the angular at a simple
straight suture. The large surangular and angular contribute to about one third of the mandible
length. The surangular, located immediately dorsal to the angular, is a large irregularly-shaped
element. It bears a low coronoid process about midway along its dorsal margin. At the
posterodorsal edge of the surangular a small cubular ossification is recognised as the articular,
although exact suture lines are hard to delineate on the specimen and latex peel. The angular
occupies the posteroventral region of the mandible. Its ventral margin is strongly curved,
especially at its posterior terminus.

DISCUSSION

***Cephalerpeton ventriarmatum*, an early eureptile from Mazon Creek**

Here we conducted the first restudy of the anatomy and systematic considerations of
early reptiles from the Pennsylvanian-aged sites of Mazon Creek, Illinois, and Linton, Ohio,
since the 1980s. Our anatomical analysis supports the placement of *Cephalerpeton*
*ventriarmatum* within Eureptilia but found characteristics in common with both Carboniferous
basal eureptiles and more derived diapsid eureptiles, such as the araeoscelids. Taking into
account that Müller and Reisz (2006) recovered *Cephalerpeton* as a member of the sister clade
(together with *Anthracoedromeus* and *Paleothyris*) to Diapsida, and that the oldest araeoscelids
occur in the Late Carboniferous (Kasimovian), it is not unreasonable to consider *Cephalerpeton*
as more closely related to diapsids than previously thought. This phylogenetic position, and the
question of whether or not *Cephalerpeton* (and by proxy, *Anthracoedromeus* and *Paleothyris*)
should be included within Diapsida, remains a possibility that needs to be tested with a
comprehensive phylogenetic analysis. We did not perform such an analysis here, as we feel it
would be preemptive in the absence of revisions of other relevant early reptiles. At present, our
understanding of the anatomy of most Pennsylvanian-aged reptiles (i.e., most members of
Protorothyrididae) is somewhat dated, and was generated when a much smaller comparative base
was available. Additionally, several recent studies have proposed major shifts in the amniote
phylogenetic tree, including the inclusion of recumbirostrans as early reptiles (Pardo et al.,
2017), varanopids as early diapsids (Ford and Benson, 2018), and some parareptiles as diapsids
(Simões et al., 2018). A new comprehensive analysis of early amniote phylogeny, synthesizing,
and testing these new hypotheses represents the appropriate first step in better understanding the
specific placement of enigmatic Carboniferous forms such as *Cephalerpeton*.

Our restudy of the anatomy of *Cephalerpeton ventriarmatum* provides not only a revised
diagnosis for the genus, but also recognises that the genus is restricted to the Mazon Creek
lagerstätte. Previously referred material from Linton, Ohio, is identified here as the new
parareptile *Carbonodraco lundi* gen. et sp. nov (formerly *Cephalerpeton* cf. aff. *C.*
*ventriarmatum*). Although we restrict *Cephalerpeton* remains at this time to only occurring in the
Mazon Creek assemblage, without the complete revision of reptile remains from Linton, Ohio,
including the closely related *Anthracoedromeus*, as well as reptile remains from other
Carboniferous sites there is no way to be certain about the level of endemism occurring in
closely-aged Carboniferous amniote communities.

*Cephalerpeton ventriarmatum* represents the oldest known reptile aside from *Hylonomus*
from Joggins, Nova Scotia, and the slightly younger *Paleothyris*, from Florence, Nova Scotia.
Unlike these earliest reptiles, which are found in lithified lycopsid tree stumps, *Cephalerpeton*
was found in a siderite nodule at Mazon Creek, which are interpreted as having been formed in
the lowland estuarine setting of a river delta (Clements et al, 2019). Traditional hypotheses of
amniote origins have described the establishment of dry, ‘upland’ ecosystems as a possible driver
of early amniote diversification (Reisz, 1972; Eberth et al, 2000); however, formal testing of this
idea has been lacking. Recently, the study of Pardo *et al.* (2019) demonstrated taxonomic
homogeneity across diverse ecosystems during the time relevant for the origin of amniotes. This
suggests, contrary to previous hypotheses, that upland ecosystems are no more abundant in early
amniotes than other ecosystems, such as the contemporaneous ‘lowland’ ecosystems, as would
be predicted if ‘upland’ ecosystems were indeed the hotbed of amniote radiation (Pardo *et al.*,
2019).

The presence of a highly terrestrial reptile within the Mazon Creek assemblage was once
thought to be an anomaly (reference). The Mazon Creek fauna is primarily dominated by aquatic
invertebrates, insects and fish (Shabica and Hay, 1978), and the preserved tetrapods found
among the Mazon Creek nodules, including the early dissorophoid non-amniote *Amphibamus*
and a number of other non-amniote taxa, are believed to have been washed in from the adjacent
near-shore environment (Clements et al., 2019). However, recent revival in the study of tetrapods
from this locality have uncovered a number of terrestrial tetrapods (e.g. recumbirostrans; Mann
et al., 2019; Mann and Maddin, 2019). Interestingly, these recumbirostrans display several
adaptations to fossoriality, suggesting the terrestrial component of Mazon Creek tetrapods has
been grossly understated. Additionally, recumbirostrans were recently recovered as a group of
reptiles (Pardo et al., 2017) revealing the estuarine paleoenvironment of the Mazon Creek
lagerstätte was likely home to a significant diversity of reptile and reptile-like terrestrial taxa.

Whereas derived terrestrial recumbirostrans from Mazon Creek are amongst the earliest
tetrapods to experiment with fossoriality, they differ in many morphological aspects from the
eureptile *Cephalerpeton*, which retains a more conventional reptilian bauplan. Despite this, there
are some important early ecological adaptations revealed by the current reanalysis of the
anatomy of *Cephalerpeton*. The most obvious of these is the dental configuration which includes
large, wide-based conical teeth, and multiple canine peaks in the maxillary tooth row, with

674 smaller recurved teeth in the dentary tooth row. Carroll and Baird (1972) and Reisz and Baird
(1983) described the maxillary dentition of *Cephalerpeton* as adequate for the processing of
hard-shelled arthropod material, which represent a widely available food source at Mazon Creek
and most Carboniferous coal measure localities (Modesto et al. 2009). We agree that the teeth
were likely used for a form of durophagous carnivory/insectivory. The wide, conical teeth with
multiple peaks along the tooth row would have done well piercing the chitinous exoskeletons of
small insects, as the lower dentition held prey in place (hence the recurvature). The dentition of
*Cephalerpeton* seems to represent an alternative approach to insectivorous durophagy
contrasting with the very wide, teardrop-shaped dentition present in durophagous gymnartrids,
pantylids, the captorhinid *Opisthodontosaurus*, and durophagous modern squamates (Carroll and
Gaskill, 1978; Reisz et al., 2015; Melstrom, 2017). The ogival tooth morphology seen in other
captorhinids is yet another example of durophagous dentition among early reptiles (Leblanc et
al., 2015). This ‘captorhinid’ type dentition could be convergently shared with CM 41714, the
indeterminate reptile from Linton, Ohio. Additionally, similar maxillary tooth morphology is
seen in both *Cephalerpeton* and the later occurring *Carbondraco*, as well as a number of
parareptiles (Modesto et al., 2009), suggesting that this durophagous tooth morphology may also
be prone to convergence.

Another overlooked aspect of *Cephalerpeton* is the length of the forelimbs, which are
proportionally longer than those present in most ‘protorothyridids’ (e.g. *Hylonomus* and
*Paleothyris*), with the exception of the closely related *Anthracodromeus*. As described above, the
length of the forelimbs more closely resembles the proportions of those of later occurring
diapsids (e.g. *Petrolacosaurus*). Carroll and Baird (1972) recognized similarities in limb
proportions and morphology of the manus and pes of the Linton taxon *Anthracodromeus* to those
of extant arboreal reptiles. Given the extensive lycopsid forests present in the Carboniferous, the
idea that *Anthracodromeus* had arboreal habits is certainly plausible. As such, it is also plausible
that *Cephalerpeton* was also adapted for some form of arboreal or at least scansorial lifestyle.

It should be noted that *Cephalerpeton* is amongst the largest of the Carboniferous
‘protorothyridid’ reptiles, only rivalled in size by *Coelostegus*. This is exceptional considering
that *Cephalerpeton* also possesses a number of features that indicate it is a juvenile (e.g. lack of
ossified ridges on the humerus). Although it is uncertain how large a mature individual of

*Cephalerpeton ventriarmatum* might have been, it would have been among the largest reptiles of
the time.

***Carbonodraco lundi* gen et sp. nov. as the oldest parareptile**

The identification of CM 23055 as a distinct taxon from *Cephalerpeton* increases the clade level
diversity of amniotes at Linton, Ohio, to include representatives of Synapsida, Eureptilia, and
now Parareptilia in the fauna. Our interpretation of the taxonomic affinity of *Carbonodraco lundi*
gen et sp. nov. further makes it the oldest known member of Parareptilia, the early diverging
sister clade to Eureptilia. Previously the oldest known parareptile, *Erpetonyx arsenaultorum*
(Modesto *et al.*, 2015), was found in the uppermost Carboniferous (Gzhelian) Egmont Bay
Formation of Prince Edward Island, Canada. *Erpetonyx* represents a very generalised parareptile
morphologically and was placed in the relatively early branching clade Bolosauria. Modesto *et*
*al.* (2015) noted that parareptiles began their evolutionary radiation before the end of the
Carboniferous Period. As a result, their time-calibrated phylogeny of parareptiles revealed long
ghost lineages for basal parareptilian clades (e.g., Mesosauridae and Millerosauria) since these
taxa appear in the fossil record in the Permian. The evolutionary framework presented in this
time-calibrated phylogeny additionally shows an Early Permian diversification of the clade
Ankyramorpha (i.e., lanthanosuchids, nyctiphuretids, and procolophonids). Recent studies by
Macdougall *et al.* (2019) on parareptile species richness and diversity through time also support
this framework. Thus our assignment of *Carbonodraco lundi* gen et sp. nov. to the **acliestorhinid**
parareptiles is of high importance to the interpretation of the timing and diversification of the
previously identified Permian-aged adaptive radiation of not only the Ankyramorpha, but the
entirety of the parareptile clades to as early as the Moscovian. The occurrence of *Carbonodraco*
*lundi* gen et sp. nov. in the Carboniferous aligns the first appearance of parareptiles close to that
of eureptiles represented by *Hylonomus* in the Bashkirian, though many long ghost lineages for
parareptiles still remain.

It then seems likely that the previously identified timing of parareptile evolution is an
artifact derived from a sampling bias of Permian-aged localities, such as Richards spur. This
revelation highlights the importance of research on new and existing tetrapod fossils from earlier,
Carboniferous-aged localities. Furthermore, phylogenetic analyses of various reptilian groups
need to be less exclusive and take into account wider taxonomic sampling from groups including

735 early eureptiles (Müller and Reisz, 2006), early diapsids (Ford and Benson, 2018) and
736 recumbirostrans (Pardo et al., 2017). Inclusions of these groups and revised anatomical analyses
have the potential to alter currently accepted evolutionary relationships of Reptilia.

**Ethics**

No ethics assessment was required prior to the completion of this research, as this study relied
entirely on museum collections. Similarly, collecting permits were not required, as no field
collections were made.

**Data accessibility**

All data, which include photographs and text description of fossils, are included in the paper.

**Authors' contributions**

748 A.M. designed the study, A.M. E.R.M E.J.M. collected and analysed the data, A.M. and
749 H.C.M. wrote the paper.

**Competing interests**

All authors declare no competing interests.

**Funding**

No funding was provided for this project.

**Acknowledgements**

We thank Amy Henrici and David S. Berman for access to specimens at the Carnegie Museum
of Natural History. We thank Diane Scott and Robert Reisz for access to parareptile material
from the Dolese Quarry, and other Permian localities. We thank John Bolt, Bob Hook, Jason
Pardo, and Bryan Gee for stimulating discussions. We further thank Bob Hook and the late
Donald Baird for access to latex peels.

REFERENCES

Brown, C. M., VanBuren, C. S., Larson, D. W., Brink, K. S., Campione, N. E., Vavrek, M. J., &
Evans, D. C. (2015). Tooth counts through growth in diapsid reptiles: implications for
interpreting individual and size-related variation in the fossil record. *Journal of Anatomy*, 226(4),
322-333.
Carroll, R. L., & Baird, D. (1972). *Carboniferous stem-reptiles of the family Romeriidae*.
Harvard University.
Clements, T., Purnell, M., & Gabbott, S. (2019). The Mazon Creek Lagerstätte: a diverse late
Paleozoic ecosystem entombed within siderite concretions. *Journal of the Geological Society*,
176(1), 1-11.
DeBraga, M., & Reisz, R. R. (1995). A new diapsid reptile from the uppermost Carboniferous
(Stephanian) of Kansas. *Palaeontology*, 38, 199-212.
Eberth, D. A., Berman, D. S., Sumida, S. S., & Hopf, H. (2000). Lower Permian terrestrial
paleoenvironments and vertebrate paleoecology of the Tambach Basin (Thuringia, central
Germany): the upland holy grail. *Palaios*, 15(4), 293-313.
Ford, D. P., & Benson, R. B. (2018). A redescription of *Orovenator mayorum* (Sauropsida,
Diapsida) using high-resolution μ CT, and the consequences for early amniote phylogeny.
*Papers in Palaeontology*.
Fox, R. C., & Bowman, M. C. (1966). Osteology and relationships of *Captorhinus aguti*
(Cope)(Reptilia: Captorhinomorpha).

Haridy, Y., Macdougall, M. J., & Reisz, R. R. (2017). The lower jaw of the Early Permian
parareptile *Delorhynchus*, first evidence of multiple denticulate coronoids in a reptile. *Zoological*
*Journal of the Linnean Society*, 184(3), 791-803.
Heaton, M. J., & Reisz, R. R. (1980). A skeletal reconstruction of the Early Permian captorhinid
reptile *Eocaptorhinus laticeps* (Williston). *Journal of Paleontology*, 136-143.
Heaton, M. J., & Reisz, R. R. (1986). Phylogenetic relationships of captorhinomorph reptiles.
*Canadian Journal of Earth Sciences*, 23(3), 402-418.
Laurin, M., & Reisz, R. R. (1995). A reevaluation of early amniote phylogeny. *Zoological*
*Journal of the Linnean Society*, 113(2), 165-223.
MacDougall, M. J., Brocklehurst, N., & Fröbisch, J. (2019). Species richness and disparity of
parareptiles across the end-Permian mass extinction. *Proceedings of the Royal Society B*,
286(1899), 20182572.
Macdougall, M. J., Scott, D., Modesto, S. P., Williams, S. A., & Reisz, R. R. (2017). New
material of the reptile *Colobomycter pholeter* (Parareptilia: Lanthanosuchoidea) and the diversity
of reptiles during the Early Permian (Cisuralian). *Zoological Journal of the Linnean Society*,
180(3), 661-671.
MacDougall, M. J., Modesto, S. P., & Reisz, R. R. (2016). A new reptile from the Richards Spur
Locality, Oklahoma, USA, and patterns of Early Permian parareptile diversification. *Journal of*
*Vertebrate Paleontology*, 36(5), e1179641.
MacDougall, M. J., Modesto, S. P., & Reisz, R. R. (2016). A new reptile from the Richards Spur
Locality, Oklahoma, USA, and patterns of Early Permian parareptile diversification. *Journal of*
*Vertebrate Paleontology*, 36(5), e1179641.

MacDougall, M. J., Tabor, N. J., Woodhead, J., Daoust, A. R., & Reisz, R. R. (2017). The unique
preservational environment of the Early Permian (Cisuralian) fossiliferous cave deposits of the
Richards Spur locality, Oklahoma. *Palaeogeography, Palaeoclimatology, Palaeoecology*, 475,
1-11.
MacDougall, M. J., LeBlanc, A. R., & Reisz, R. R. (2014). Plicidentine in the Early Permian
parareptile *Colobomycter pholeter*, and its phylogenetic and functional significance among
coeval members of the clade. *PloS One*, 9(5), e96559.
Modesto, S. P., Scott, D. M., MacDougall, M. J., Sues, H. D., Evans, D. C., & Reisz, R. R.
(2015). The oldest parareptile and the early diversification of reptiles. *Proceedings of the Royal*
*Society B: Biological Sciences*, 282(1801), 20141912.
Modesto, S. P., Scott, D. M., & Reisz, R. R. (2009). Arthropod remains in the oral cavities of
fossil reptiles support inference of early insectivory. *Biology Letters*, 5(6), 838-840.
Modesto, S. P. (1999). *Colobomycter pholeter* from the Lower Permian of Oklahoma: a
parareptile, not a protorothyridid. *Journal of Vertebrate Paleontology*, 19(3), 466-472.
Modesto, S. P., & Reisz, R. R. (2008). New material of *Colobomycter pholeter*, a small
parareptile from the Lower Permian of Oklahoma. *Journal of Vertebrate Paleontology*, 28(3),
677-684.
Moodie, R. L. (1912). *The Pennsylvanian Amphibia of the Mazon Creek, Illinois, Shales*.
University of Kansas.
Moodie, R. L. (1916). *The coal measures amphibia of North America* (No. 238). Carnegie
Institution of Washington.

Müller, J., Berman, D. S., Henrici, A. C., Martens, T., & Sumida, S. S. (2006). The basal reptile
*Thuringothyris mahlendorffae* (Amniota: Eureptilia) from the Lower Permian of Germany.
*Journal of Paleontology*, 80(4), 726-739.
Müller, J., & Reisz, R. R. (2006). The phylogeny of early eureptiles: comparing parsimony and
Bayesian approaches in the investigation of a basal fossil clade. *Systematic biology*, 55(3), 503-
511.
Pardo, J. D., Szostakiwskyj, M., Ahlberg, P. E., & Anderson, J. S. (2017). Hidden morphological
diversity among early tetrapods. *Nature*, 546(7660), 642.
Reisz, R., & Baird, D. (1983). *Captorhinomorph" stem" reptiles from the Pennsylvanian coal-*
*swamp deposit of Linton, Ohio*. Carnegie Museum of Natural History.
Reisz, R. R. (1977). *Petrolacosaurus*, the oldest known diapsid reptile. *Science*, 196(4294), 1091-
1093.
Reisz, R. (1972). *Pelycosaurian reptiles from the middle Pennsylvanian of North America*.
Harvard University.
Reisz, R. R., LeBlanc, A. R., Sidor, C. A., Scott, D., & May, W. (2015). A new captorhinid
reptile from the Lower Permian of Oklahoma showing remarkable dental and mandibular
convergence with microsaurian tetrapods. *The Science of Nature*, 102(9-10), 50.
Reisz, R., Haridy, Y., & Müller, J. (2016). *Euconcordia* nom. nov., a replacement name for the
captorhinid eureptile *Concordia* Müller and Reisz, 2005 (non Kingsley, 1880), with new data on
its dentition. *Vertebrate Anatomy Morphology Palaeontology*, 3.
Reisz, R. R., Berman, D. S., & Scott, D. (1984). The anatomy and relationships of the Lower
Permian reptile *Araeoscelis*. *Journal of Vertebrate Paleontology*, 4(1), 57-67.

Shabica, C. W., & Hay, A. (Eds.). (1997). Richardson's guide to the fossil fauna of Mazon
Creek. Northeastern Illinois University.

Simões, T. R., Caldwell M. W., Nydam R. L. (2018). Re-grafting early reptile phylogeny by
increased taxon sampling and multiple optimality criteria. In *6th Annual Meeting Canadian*
*Society of Vertebrate Palaeontology May 14-16, 2018 Ottawa, Ontario*(p. 7).

**Figure captions**

**Figure 1:** *Cephalerpeton ventriarmatum* (YPM 796), A) photograph of the original nodule, and
B) a resin cast of the original nodule.

**Figure 2:** *Cephalerpeton ventriarmatum* (YPM 796), A) latex peel and B) illustration of the
cranial and postcranial anatomy, modified from Carroll and Baird (1972). C) Cranial
reconstruction of *Cephalerpeton ventriarmatum*.

**Figure 3:** Photographs of *Carbonodraco lundi* gen. et sp. nov., A) the holotype specimen (CM
23055) and B) latex peel of the holotype specimen.

**Figure 4:** Illustration of the cranial anatomy of the holotype of *Carbonodraco lundi* gen. et sp
nov. (CM 23055), and a reconstruction.

**Figure 5:** Referred specimens of *Carbonodraco lundi* gen. et sp. nov. A) Latex peel of NHMUK
R. 2667 (J. W. Davies Linton Collection) showing a right mandible in lingual perspective. B-D)
Specimen CM 81536. B) Original cancell coal specimen, C) latex peel, and D) interpretive
drawing, showing a pair of dentaries preserved in lingual perspective (collected by Scott
Mckenzie).

**Figure 6:** Indeterminate reptilian amniote from Linton, Ohio (CM 41714). A) Latex peel, B)
interpretive drawing, and C) original cancell coal specimen.

Figure 1: *Cephaleperon ventriarmatum* (YPM 796), A) photograph of the original nodule, and B) a resin cast of the original nodule.

182x101mm (300 x 300 DPI)

Figure 2: *Cephalepeton ventriarmatum* (YPM 796), A) latex peel and B) illustration of the cranial and postcranial anatomy, modified from Carroll and Baird (1972). C) Cranial reconstruction of *Cephalepeton ventriarmatum*.

181x149mm (300 x 300 DPI)

Figure 3: Photographs of *Carbonodraco lundii* gen. et sp. nov., A) the holotype specimen (CM 23055) and B) latex peel of the holotype specimen.

181x229mm (300 x 300 DPI)

Figure 4: Illustration of the cranial anatomy of the holotype of *Carbonodraco lundii* gen. et sp. nov. (CM 23055), and a reconstruction.

Figure 5: Referred specimens of *Carbonodraco lundii* gen. et sp. nov. A) Latex peel of NHMUK R. 2667 (J. W. Davies Linton Collection) showing a right mandible in lingual perspective. B-D) Specimen CM 81536. B) Original cannel coal specimen, C) latex peel, and D) interpretive drawing, showing a pair of dentaries preserved in lingual perspective (collected by Scott Mckenzie).

181x149mm (300 x 300 DPI)

Figure 6: Indeterminate reptilian amniote from Linton, Ohio (CM 41714). A) Latex peel, B) interpretive drawing, and C) original cancellal coal specimen.

181x149mm (300 x 300 DPI)

Appendix B

Reviewer responses.

Dear editorial staff,

Please find my responses to the reviewers comments, since the reviewers comments were minor, most were accepted with rationale provided in a few instances where they were not.

Additionally, all changes requested in the pdf document (by reviewer 2) were made.

Reviewer comments to Author:

Reviewer: 1

Comments to the Author(s)

This is a nice little paper that clarifies the amniote presence at the Carboniferous of the U.S.A. After minor revisions, it would be eminently suitable for publication in Royal Society Open Science. I think the paper would be greatly enhanced by an updated list(s) of the vertebrate/tetrapod fauna now recognized at the Mazon Creek and Linton sites.

Response: The first suggestion here to provide an updated taxonomic list is a good one, but has to wait until more faunal constituents are revised. At present the current taxonomic lists do not deviate from those published previously by Hook and Baird (1986; 1988) for Linton, or Shabica and Hay (1997) for Mazon Creek. This is my eventual goal, to rework these fauna, currently my almost complete thesis on Mazon Creek tetrapoda, will provide an updated list in a future publication for that site first.

I have the following minor comments/corrections (based on the pagination in the Word document):

Line 46: typo in 'protorthyridids' (here and elsewhere in text)

Response: changed

Line 53: typo in 'acliestorhinids' (here and elsewhere in text, e.g. line 442)

Response: changed

Line 64: the authors use the term 'sauropsids' whereas they use the term 'reptiles' in the title and elsewhere in their paper; it is not clear if the authors are using these terms interchangeably or if they refer to different groupings of amniotes; if the former, the authors should pick one term and use it consistently

Response: changed to reptiles.

Line 70: the locality name 'Nyřany' is missing diacritical marks [Nýřany]

Response: changed

Line 73: given my comments for line 64, I am not sure what the authors mean by 'stem-reptile family': do they mean that protorothyridids are stem reptiles sensu Robert Carroll? Or do they mean that they regard Reptilia to be a crown group and thus Sauropsida is the total group? Please clarify.

Response: Stem-reptile removed.

Line 118: typo in 'Acliestorhinus'

Response: changed

Line 129: there are two abbreviations for 'exoccipital' ('eo' is used in figure 2b)

Response: eo retained only.

Line 164: 'Macedougall' should be 'MacDougall' (here and elsewhere in text)

Response: changed

Line 178: one can estimate skull length, or measure the exact length of a skull, but not 'estimate' the exact skull length

Response: exact removed, language changed.

Line 306: change 'cuboidal' to 'quadrangular'

Response: changed

Line 311: each gnathostome has a single mandible (there is no such thing as a right mandible in tetrapod biology, unless you are describing a teratology); see also lines 435 and 909

Response: changed throughout text to lower jaw where applicable.

Line 314: delete one of the full stops following 'nov'. See also lines 547, 589

Response: changed

Line 322: change 'however, no sutures can be discerned' to 'but no sutures can be discerned'

Response: changed

Line 330: change 'which is likely accurate' to 'with which we agree'

Response: changed

Line 421: excuse my pedantry, but the Latin 'draco' is from ancient Greek 'drakon', which means 'serpent' and was used to refer to snakes of unusual size ('ophis' was used for small snakes). That is to say, the ancient Romans and Greeks did not associate their terms with (mythological) winged reptiles, as this is a later concept (the English 'dragon' is derived from 'draco'). I am not suggesting a name change for the authors' new genus, but they should be clear on the etymology; authors may wish to check out the book "Dragon Myth and Serpent Cult in the Greek and Roman Worlds" by Daniel Ogden for background.

Response: Changed to serpent after the latin.

Line 436: typo in 'Riesz'

Response: fixed.

Line 449: change 'are not able to be identified at present' to 'are not identifiable at present'

Response: changed.

Line 461: change 'recurvatures' to 'recurved'

Response: changed.

Line 657: a reference is needed here

Response: added.

Line 670: I have worked on extinct reptiles for many years, and I find the phrase 'more conventional reptilian bauplan' puzzling. Please elaborate on this concept.

Response: Changed to stereotypical lizard-like bauplan

Line 781: author name is 'deBraga'

Response: changed.

Lines 790, 793, and 800: information missing from these references

Response: added, and removed ref to Fox and Bowman.

Figure 2: label C missing from skull reconstruction

Response: figure re-arranged- latex cast removed.

Reviewer: 2

Comments to the Author(s)

This is an interesting paper that extends the lineage of one important clade of Permian reptiles. The authors have nicely demonstrated that Carbonodraco is a parareptile.

I request the authors to use the term Reptilia, this name clearly has priority over the unnecessary "Sauropsida". See Modesto and Anderson (2004) for a phylogenetic definition of Reptilia.

Response: done!

I made some minor comments and corrected some typos (see attached file). The names *Acleistorhinus* and *acleistorhinids* are misspelled throughout the entire text. Please also check my comments to the illustrations in that file.

Response: all changes made!

Reviewer: 3

Comments to the Author(s)

This paper re-describes some interesting specimens representing early eureptilians and parareptiles, which are important for understanding the origin and evolution of Reptilia. The descriptions are thorough, and my comments mainly refer to the terminology used to describe the teeth:

Line 99: "crafted by Donald Baird" sounds like he made it up. Maybe say "... we were able to study original latex peels taken/collected/made by DB and casts of..."

Response: changed to made by.

Line 156: "maxilla bears multiple canine-like peaks". What does this mean? Are all the teeth caniniform in shape? Using the word 'teeth' instead of peaks would make much more sense here, if this statement is actually referring to teeth and not some peaks made of bone on the maxilla. If this statement is referring to tooth size, maybe say: Maxillary teeth irregular in size along toothrow/ maxillary tooth height inconsistent along toothrow/ or something like that. See comments below regarding the use of 'canine' and separating tooth position, tooth size, and tooth shape in descriptions.

Response: changed to 'maxillary dentition bears multiple caniniform peaks'

Line 188: "There a thin and delicately built anatomy is seen..." Add a comma after "There".

Response: done.

Line 195: The lacrimal

Response: added the.

Line 199: The word "canines" in parentheses is not necessary here, since canine refers to a specific tooth type and tooth position in mammals. See comments below.

Response: removed.

Line 215: Sides missing an S.

Response: added

Line 320: confusing sentence "On the right, lingually preserved, dentary the symphysis..."

Response: reworded to 'On the right dentary, preserved in lingual aspect, the symphysis....'

Line 444, 460, 462, 474: the use of 'canine' and 'incisor' is problematic, as these are mammalian terms referring to a specific location in the mouth and specific tooth shapes. Check out the discussion in Macdougall and Reisz 2014 on the use of the term 'canine' and 'caniniform' and switch your language accordingly (talk about size differences in term of tooth number, e.g., maxillary teeth 4 and 5 are twice the length of all other maxillary teeth). This will make scoring these animals in future phylogenetic analyses easier. Separate tooth size from tooth shape and tooth position.

Response: Language has been changed in accordance with the reviewer remarks, Incisor and canine have been removed and replaced with enlarged teeth respectively.

MacDougall, M. J., and R. R. Reisz. 2014. The first record of a nyctiphruetid parareptile from the Early Permian of North America, with a discussion of parareptilian temporal fenestration. *Zoological Journal of the Linnean Society* 172:616-630.

Line 449: "are not able to be identified at present" can be "are not identifiable at present"

Response: done.

Line 461: "gently recurvated" should be gently recurved

Response: done.

Line 542: '...have places of 19 teeth...' should be 'for' instead of 'of'

Response: done.

Line 588: '... narrowing morphology that end with a...' missing s

Response: added.

Line 650: Pardo et al. (2019) missing from reference list.

Response: added.

Lines 651-655 are very confusing and should be re-written for clarity, especially 'Upland ecosystems are no more abundant in early amniotes than other ecosystems'

Response: rearranged, to 'Traditional hypotheses of amniote origins have described the establishment of dry, 'upland' ecosystems as a possible driver of early amniote diversification (Reisz, 1972; Eberth et al, 2000); however, formal testing of this idea has been lacking until recently (see Pardo et al. 2019). The presence of a highly terrestrial amniote fauna (including synapsids, eureptiles and parareptiles) at the 'low-land' localities of Mazon Creek and Linton also suggests a more complicated evolutionary scenario for early amniote diversification.'

Line 657 (reference)

Response: section removed.

Line 673, 679 'multiple canine peaks in the maxillary tooth row'. See comments above. Separate tooth shape from tooth size and tooth position.

Response: changed.

Line 678-680: Can you cite something that you would support your interpretation of this type of feeding?

Response: cited Melstrom 2017 and added to ref list.

Line 698: 'had arboreal habits' should be 'was arboreal'

Response: changed.

Figure 2: The peel doesn't offer much information. Is it possible to make the drawing bigger in order to better see the described anatomy? This would make the description much easier

to follow (especially the palate). The scleral ossicles should be included in the cranial reconstruction.

Response: changed.

Figure 3: This figure doesn't show anything informative. Maybe combine with figure 4, but maintain the detail and size of the interpretive line drawing? Consider putting photos of peels as supp info and just focus on the detailed drawings for the figures in the paper?

Response: unchanged, I believe it necessary to show the raw data for this animal in the paper, and what the specimen actually looks like, it has never been imaged before either.

References: most species names missing italics

Response: changed.